# Evolution of a plant gene cluster in Solanaceae and emergence of metabolic diversity

Pengxiang Fan[1†], Peipei Wang[2], Yann-Ru Lou[1], Bryan J Leong[2], Bethany M Moore[2,3], Craig A Schenck[1], Rachel Combs[4‡], Pengfei Cao[2,5], Federica Brandizzi[2,5], Shin-Han Shiu[2,6], Robert L Last[1,2]*

[1]Department of Biochemistry and Molecular Biology, Michigan State University, East Lansing, United States; [2]Department of Plant Biology, Michigan State University, East Lansing, United States; [3]University of Wisconsin, Madison, United States; [4]Division of Biological Sciences, University of Missouri, Columbus, United States; [5]MSU-DOE Plant Research Laboratory, Michigan State University, East Lansing, United States; [6]Department of Computational Mathematics, Science, and Engineering, Michigan State University, East Lansing, United States

*For correspondence:
lastr@msu.edu

Present address: [†]College of Agriculture and Biotechnology, Zijingang Campus, Zhejiang University, Hangzhou, People's Republic of China; [‡]Center for Applied Plant Sciences, The Ohio State University, Columbus, United States

Competing interests: The authors declare that no competing interests exist.

**Abstract** Plants produce phylogenetically and spatially restricted, as well as structurally diverse specialized metabolites via multistep metabolic pathways. Hallmarks of specialized metabolic evolution include enzymatic promiscuity and recruitment of primary metabolic enzymes and examples of genomic clustering of pathway genes. Solanaceae glandular trichomes produce defensive acylsugars, with sidechains that vary in length across the family. We describe a tomato gene cluster on chromosome 7 involved in medium chain acylsugar accumulation due to trichome specific acyl-CoA synthetase and enoyl-CoA hydratase genes. This cluster co-localizes with a tomato steroidal alkaloid gene cluster and is syntenic to a chromosome 12 region containing another acylsugar pathway gene. We reconstructed the evolutionary events leading to this gene cluster and found that its phylogenetic distribution correlates with medium chain acylsugar accumulation across the Solanaceae. This work reveals insights into the dynamics behind gene cluster evolution and cell-type specific metabolite diversity.

## Introduction

Despite the enormous structural diversity of plant specialized metabolites, they are derived from a relatively small number of primary metabolites, such as sugars, amino acids, nucleotides, and fatty acids (*Maeda, 2019*). These lineage-, tissue- or cell- type specific specialized metabolites mediate environmental interactions, such as herbivore and pathogen deterrence or pollinator and symbiont attraction (*Mithöfer and Boland, 2012*; *Pichersky and Lewinsohn, 2011*). Specialized metabolism evolution is primarily driven by gene duplication (*Moghe and Last, 2015*; *Panchy et al., 2016*), and relaxed selection of the resulting gene pairs allows modification of cell- and tissue-specific gene expression and changes in enzymatic activity. This results in expanded substrate recognition and/or diversified product formation (*Khersonsky and Tawfik, 2010*; *Leong and Last, 2017*). The neofunctionalized enzymes can prime the origin and diversification of specialized metabolic pathways (*Schenck and Last, 2020*; *Weng et al., 2012*; *Weng, 2014*).

There are many examples of mechanisms that lead to novel enzymatic activities in specialized cell- or tissue-types, however, the principles that govern assembly of multi-enzyme specialized metabolic pathways are less well established. One appealing hypothesis involves the stepwise recruitment of pathway enzymes (*Noda-Garcia et al., 2018*). In rare cases, non-homologous specialized

**eLife digest** Plants produce a vast variety of different molecules known as secondary or specialized metabolites to attract pollinating insects, such as bees, or protect themselves against herbivores and pests. The secondary metabolites are made from simple building blocks that are readily available in plants, including amino acids, fatty acids and sugars.

Different species of plant, and even different parts of the same plant, produce their own sets of secondary metabolites. For example, the hairs on the surface of tomatoes and other members of the nightshade family of plants make metabolites known as acylsugars. These chemicals deter herbivores and pests from damaging the plants.

To make acylsugars, the plants attach long chains known as fatty acyl groups to molecules of sugar, such as sucrose. Some members of the nightshade family produce acylsugars with longer chains than others. In particular, acylsugars with long chains are only found in tomatoes and other closely-related species. It remained unclear how the nightshade family evolved to produce acylsugars with chains of different lengths.

To address this question, Fan et al. used genetic and biochemical approaches to study tomato plants and other members of the nightshade family. The experiments identified two genes known as *AACS* and *AECH* in tomatoes that produce acylsugars with long chains. These two genes originated from the genes of older enzymes that metabolize fatty acids – the building blocks of fats – in plant cells. Unlike the older genes, *AACS* and *AECH* were only active at the tips of the hairs on the plant's surface. Fan et al. then investigated the evolutionary relationship between 11 members of the nightshade family and two other plant species. This revealed that *AACS* and *AECH* emerged in the nightshade family around the same time that longer chains of acylsugars started appearing.

These findings provide insights into how plants evolved to be able to produce a variety of secondary metabolites that may protect them from a broader range of pests. The gene cluster identified in this work could be used to engineer other species of crop plants to start producing acylsugars as natural pesticides.

metabolic enzyme genes occur in proximity to each other in a genomic region, forming a biosynthetic gene cluster (*Nützmann et al., 2016*; *Nützmann and Osbourn, 2014*; *Rokas et al., 2018*). In recent years, an increasing number of specialized metabolic gene clusters (SMGCs) were experimentally identified or bioinformatically predicted in plants (*Boutanaev et al., 2015*; *Castillo et al., 2013*; *Schläpfer et al., 2017*). However, although most experimentally characterized plant SMGCs are co-expressed, the majority of the bioinformatically predicted ones do not show coexpression under global network analysis (*Wisecaver et al., 2017*).

While examples of SMGCs are still relatively rare in plants, experimentally validated cases were reported for a surprisingly diverse group of pathways. These include terpenes (*Chae et al., 2014*; *Prisic et al., 2004*; *Qi et al., 2004*; *Wilderman et al., 2004*), cyclic hydroxamic acids (*Frey et al., 1997*), biosynthetically unrelated alkaloids (*Itkin et al., 2013*; *Winzer et al., 2012*), polyketides (*Schneider et al., 2016*), cyanogenic glucosides (*Takos et al., 2011*), and modified fatty acids (*Jeon et al., 2020*). However, whereas each cluster encodes multiple non-homologous enzymes of a biosynthetic pathway, evolution of their assembly is not well understood.

Acylsugars are a group of insecticidal (*Leckie et al., 2016*) and anti-inflammatory (*Herrera-Salgado et al., 2005*) chemicals mainly observed in glandular trichomes of Solanaceae species (*Fan et al., 2019*; *Schuurink and Tissier, 2020*). These specialized metabolites are sugar aliphatic esters with three levels of structural diversity across the Solanaceae family: acyl chain length, acylation position, and sugar core (*Fan et al., 2019*). The primary metabolites sucrose and aliphatic acyl-CoAs are the biosynthetic precursors of acylsucroses in plants as evolutionarily divergent as the cultivated tomato *Solanum lycopersicum* (*Fan et al., 2016*; *Figure 1*), *Petunia axillaris* (*Nadakuduti et al., 2017*) and *Salpiglossis sinuata* (*Moghe et al., 2017*). The core tomato acylsucrose biosynthetic pathway involves four BAHD [**B**EAT, **A**HCT, **H**CBT, **D**AT (*D'Auria, 2006*) family acylsucrose acyltransferases (*Sl-ASAT1* through *Sl-ASAT4*), which are specifically expressed in the type I/IV trichome tip cells (*Fan et al., 2016*; *Schilmiller et al., 2015*; *Schilmiller et al., 2012*). These

**Figure 1.** Primary metabolites are biosynthetic precursors of tomato trichome acylsugars. In cultivated tomatoes, the trichome acylsucroses are synthesized by four Sl-ASATs using the primary metabolites – sucrose and different types of acyl-CoAs – as substrates. In this study we provide evidence that medium chain fatty acids are converted to acyl-CoAs by an acyl-CoA synthetase for medium chain acylsugar biosynthesis.

enzymes catalyze consecutive reactions utilizing sucrose and acyl-CoA substrates to produce the full set of cultivated tomato acylsucroses in vitro (*Fan et al., 2016*).

Co-option of primary metabolic enzymes contributed to the evolution of acylsugar biosynthesis and led to interspecific structural diversification across the *Solanum* tomato clade. One example is an invertase-like enzyme originating from carbohydrate metabolism that generates acylglucoses in the wild tomato *S. pennellii* through cleavage of the acylsucrose glycosidic bond (*Leong et al., 2019*). In another case, allelic variation of a truncated isopropylmalate synthase-like enzyme (IPMS3) – from branched chain amino acid metabolism – leads to acylsugar iC4/iC5 (2-methylpropanoic/3-methylbutanoic acid) acyl chain diversity in *S. pennellii* and *S. lycopersicum* (*Ning et al., 2015*). Acylsugar structural diversity is even more striking across the family. Previous studies revealed variation in acyl chain length (*Ghosh et al., 2014*; *Liu et al., 2017*; *Moghe et al., 2017*): *Nicotiana*, *Petunia* and *Salpiglossis* species were reported to accumulate acylsugars containing only short acyl chains (carbon number, $C \leq 8$). In contrast, some species in *Solanum* and other closely related genera produce acylsugars with medium acyl chains ($C \geq 10$). These results are consistent with the hypothesis that the capability to produce medium chain acylsugars varies across the Solanaceae family.

In this study, we identify a metabolic gene cluster on tomato chromosome 7 containing two non-homologous genes – acylsugar acyl-CoA synthetase (*AACS*) and acylsugar enoyl-CoA hydratase (*AECH*) – affecting medium chain acylsugar biosynthesis. Genetic and biochemical results show that the trichome enriched *AACS* and *AECH* are involved in generating medium chain acyl-CoAs, which are donor substrates for acylsugar biosynthesis. Genomic analysis revealed a syntenic region on chromosome 12, where the acylsucrose biosynthetic *Sl-ASAT1* is located (*Fan et al., 2016*). Phylogenetic analysis of the syntenic regions in Solanaceae and beyond led to evolutionary reconstruction of the origin of the acylsugar gene cluster. We infer that sequential gene insertion facilitated emergence of this gene cluster in tomato. These results provide insights into specialized metabolic evolution through emergence of cell-type specific gene expression, the formation of metabolic gene clusters and illuminates additional examples of primary metabolic enzymes being co-opted into specialized metabolism.

## Results

### Identification of a metabolic gene cluster that affects tomato trichome medium chain acylsugar biosynthesis

*S. pennellii* natural accessions (*Mandal et al., 2020*), as well as the *S. lycopersicum* M82 $\times$ *S. pennellii* LA0716 chromosomal substitution introgression lines (ILs) (*Eshed and Zamir, 1995*), offer convenient

resources to investigate interspecific genetic variation that affects acylsugar metabolic diversity (*Mandal et al., 2020*; *Schilmiller et al., 2010*). In a rescreen of ILs for *S. pennellii* genetic regions that alter trichome acylsugar profiles (*Schilmiller et al., 2010*), IL7-4 was found to accumulate increased C10 medium chain containing acylsugars compared with M82 (*Figure 2*, A and B). The genetic locus that contributes to the acylsugar phenotype was narrowed down to a 685 kb region through screening selected backcross inbred lines (BILs) (*Ofner et al., 2016*) that have recombination breakpoints on chromosome 7 (*Figure 2C*). Because tomato acylsucrose biosynthesis occurs in trichomes, candidate genes in this region were filtered based on their trichome-specific expression patterns. This analysis identified a locus containing multiple tandemly duplicated genes of three families – an acyl-CoA synthetase (ACS), enoyl-CoA hydratase (ECH), and BAHD acyltransferase. Our analysis (*Moore et al., 2020*) revealed co-expression of four *Sl-ASATs* (*Fan et al., 2019*) and three genes at the locus – *Solyc07g043630*, *Solyc07g043660*, and *Solyc07g043680* (*Supplementary file 1*

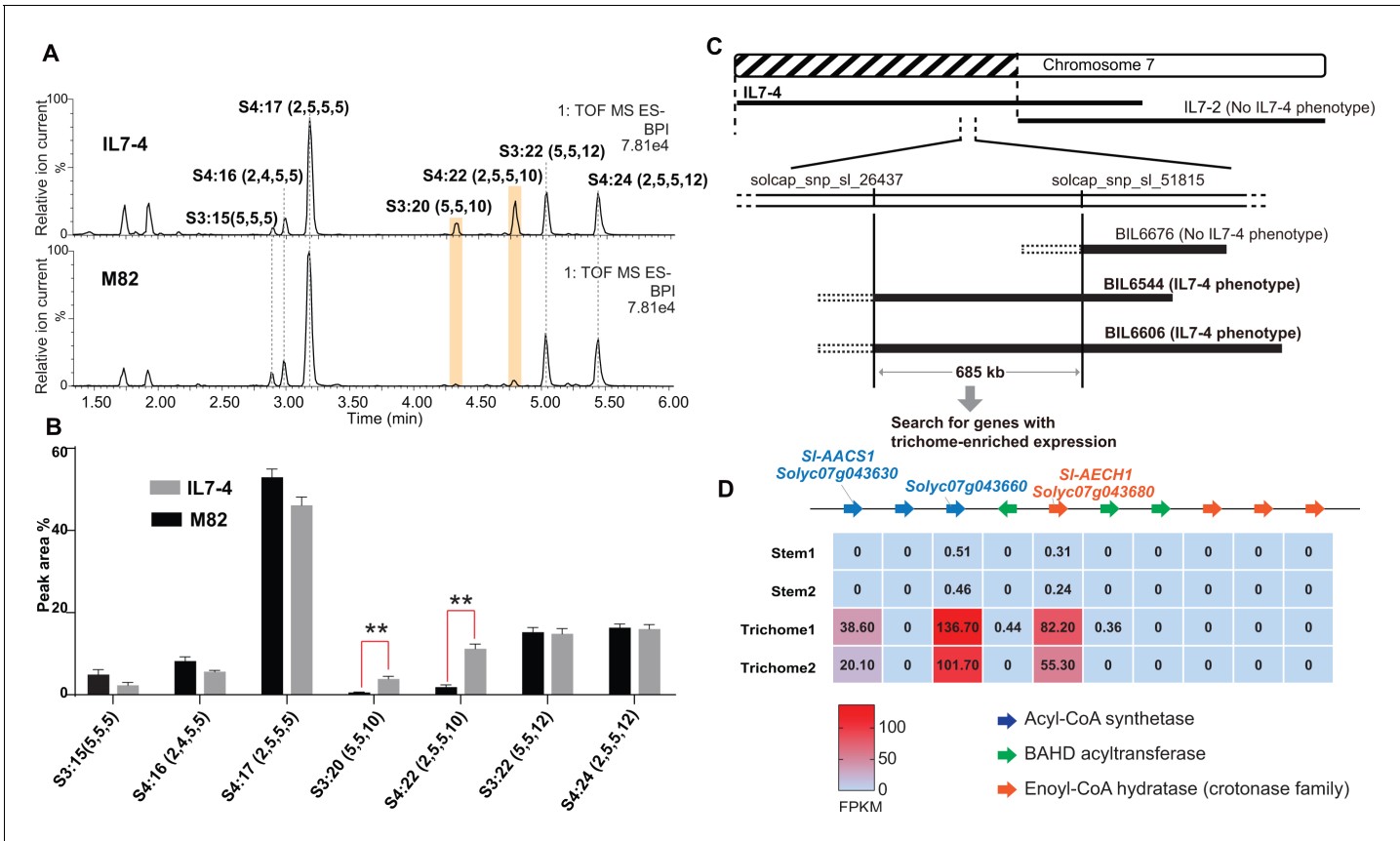

**Figure 2.** Mapping of a genetic locus related to acylsugar variations in tomato interspecific introgression lines. (A) Electrospray ionization negative (ESI⁻) mode, base-peak intensity (BPI) LC/MS chromatogram of trichome metabolites from cultivated tomato *S. lycopersicum* M82 and introgression line IL7-4. The orange bars highlight two acylsugars that have higher abundance in IL7-4 than in M82. For the acylsucrose nomenclature, 'S' refers to a sucrose backbone, '3:22' means three acyl chains with twenty-two carbons in total. The length of each acyl chain is shown in the parentheses. (B) Peak area percentage of seven major trichome acylsugars in M82 and IL7-4. The sum of the peak area percentage of each acylsugar is equal to 100% in each sample. The data is shown for three plants ± SEM. \*\*p<0.01, Welch two-sample *t* test. *Figure 2—source data 1* includes values for the analysis. (C) Mapping the genetic locus contributing to the IL7-4 acylsugar phenotype using selected backcross inbred lines (BILs) that have recombination break points within the introgression region of IL7-4. (D) Narrowing down candidate genes in the locus using trichome/stem RNA-seq datasets generated from previous study (*Ning et al., 2015*). A region with duplicated genes of three types – acyl-CoA synthetase (ACS), BAHD acyltransferase, and enoyl-CoA hydratase (ECH) – is shown. The red-blue color gradient provides a visual marker to rank the expression levels represented by Fragments Per Kilobase of transcript per Million mapped reads (FPKM). Coexpression analysis of tomato ACS, ECH, and BAHD acyltransferase family genes is shown in *Figure 2—figure supplement 1*.

The online version of this article includes the following source data and figure supplement(s) for figure 2:

**Source data 1.** Data used to make *Figure 2B*.

**Figure supplement 1.** Expression profiles of tomato ACS, ECH, and BAHD acyltransferase family genes used for phylogenetic analysis in this study.

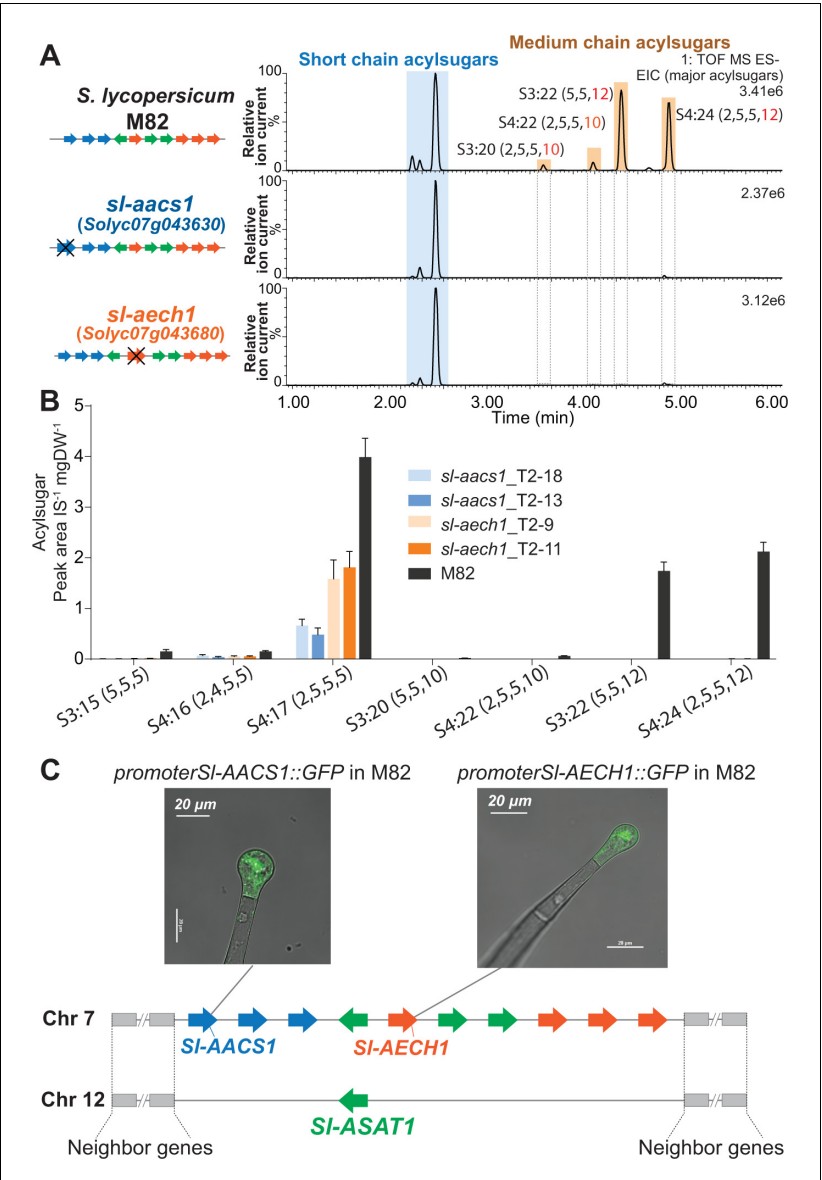

**Figure 3.** CRISPR/Cas9-mediated gene knockout of tomato *Sl-AACS1* or *Sl-AECH1* eliminates detectable medium chain containing acylsugars. (**A**) Combined LC/MS extracted ion chromatograms of trichome metabolites from CRISPR mutants *sl-aacs1* and *sl-aech1*. The medium chain acylsugars that are not detected in the two mutants are denoted by pairs of vertical dotted lines. *Figure 3—figure supplement 1* describes the design of the gRNAs and details of the gene edits. (**B**) Quantification of seven major trichome acylsugars in *sl-aacs1* and *sl-aech1* mutants. Two independent T2 generation transgenic lines for each mutant were used for analysis. The peak area/internal standard (IS) normalized by leaf dry weight (DW) is shown from six plants ± SEM. *Figure 3—source data 1* includes values for the analysis. (**C**) Confocal fluorescence images showing that GFP fluorescence driven by *Sl-AACS1* or *Sl-AECH1* is located in the tip cells of type I/IV trichomes. Their tissue specific expressions are similar to *Sl-ASAT1* (*Fan et al., 2016*), which locates in a chromosome 12 region that is syntenic to the locus containing *Sl-AACS1* and *Sl-AECH1*. *Figure 3—figure supplement 2* provides the detailed information of the syntenic region. *Sl-AACS1*, *Sl-AECH1*, and *Sl-ASAT1* are the only gene models with demonstrated functions in acylsugar biosynthesis.

The online version of this article includes the following source data and figure supplement(s) for figure 3:

**Source data 1.** Data used to make *Figure 3B*.
**Figure supplement 1.** CRISPR-Cas9-mediated gene knockouts in cultivated tomato *S. lycopersicum*.
**Figure supplement 2.** The syntenic region of cultivated tomato *S. lycopersicum* chromosome 7 and 12 harboring the acylsugar and steroidal glycoalkaloid gene clusters.

and *Figure 2—figure supplement 1*). Expression of these three genes was trichome enriched (*Figure 2D*), and thus they were selected for further analysis.

The three candidate genes were tested for involvement in tomato acylsugar biosynthesis by making loss of function mutations using the CRISPR-Cas9 gene editing system. Two guide RNAs (gRNAs) were designed to target one or two exons of each gene to assist site-specific DNA cleavage by hCas (*Brooks et al., 2014*; *Figure 3—figure supplement 1, A–C*). In the self-crossed T1 progeny of stably transformed M82 plants, at least two homozygous mutants were obtained in *Solyc07g043630*, *Solyc07g043660*, and *Solyc07g043680* (*Figure 3—figure supplement 1, A–C*), and these were analyzed for leaf trichome acylsugar changes. Altered acylsugar profiles were observed in the ACS-annotated *Solyc07g043630* or ECH-annotated *Solyc07g043680* mutants (*Figure 3*, A and B), but not in the ACS-annotated *Solyc07g043660* mutant (*Figure 3—figure supplement 1D*). Despite carrying mutations in distinctly annotated genes (ACS or ECH), the two mutants exhibited the same phenotype – no detectable medium acyl chain (C10 or C12) containing acylsugars (*Figure 3*, A and B). We renamed *Solyc07g043630* as <u>a</u>cylsugar <u>a</u>cyl-<u>C</u>oA <u>s</u>ynthetase 1 (*Sl-AACS1*) and *Solyc07g043680* as <u>a</u>cylsugar <u>e</u>noyl-<u>C</u>oA <u>h</u>ydratase 1 (*Sl-AECH1*) based on this analysis.

Further genomic analysis revealed that *Sl-AACS1* and *Sl-AECH1* belong to a syntenic region shared with a locus on chromosome 12, where *Sl-ASAT1* is located (*Figure 3C* and *Figure 3—figure supplement 2*). *Sl-ASAT1* is specifically expressed in trichome tip cells and encodes the enzyme catalyzing the first step of tomato acylsucrose biosynthesis (*Fan et al., 2016*). This led us to test the cell-type expression pattern of *Sl-AACS1* and *Sl-AECH1*. Like *Sl-ASAT1*, the promoters of both genes drove GFP expression in the trichome tip cells of stably transformed M82 plants (*Figure 3C*). This supports our hypothesis that *Sl-AACS1* and *Sl-AECH1* are involved in tomato trichome acylsugar biosynthesis. Taken together, we identified a metabolic gene cluster involved in medium chain acylsugar biosynthesis, which is composed of two cell-type specific genes.

## In vitro analysis of Sl-AACS1 and Sl-AECH1 implicates their roles in medium chain acyl-CoA metabolism

ACS and ECH are established to function in multiple cell compartments for the metabolism of acyl-CoA (*Buchanan et al., 2015*), the acyl donor substrates for ASAT enzymes. We sought to understand the organelle targeting of Sl-AACS1 and Sl-AECH1, to advance our knowledge of acylsugar machinery at the subcellular level. We constructed expression cassettes of Sl-AACS1, Sl-AECH1 and Solyc07g043660 with C-terminal cyan fluorescent protein (CFP), hypothesizing that the targeting peptides reside at the N-terminus of precursor proteins. When co-expressed in tobacco leaf epidermal cells, three CFP-tagged recombinant proteins co-localized with the mitochondrial marker MT-RFP (*Nelson et al., 2007*; *Figure 4A* and *Figure 4—figure supplement 1A*). To rule out the possibility of peroxisomal localization, we fused Sl-AACS1, Sl-AECH1, or Solyc07g043660 with N-terminus fused yellow fluorescent protein (YFP), considering that potential peroxisomal targeting peptides are usually located on the C-terminus (*Brocard and Hartig, 2006*). The expressed YFP-recombinant proteins were not co-localized with the peroxisomal marker RFP-PTS (*Nelson et al., 2007*; *Figure 4—figure supplement 1B*). Instead, they appeared distributed in the cytosol (*Figure 4—figure supplement 1B*), presumably because the N-terminal YFP blocked the mitochondria targeting signal. Taken together, protein expression and co-localization analyses suggest that *Sl-AACS1*, *Sl-AECH1*, and *Solyc07g043660* encode enzymes targeted to mitochondria.

Sl-AACS1 belongs to a group of enzymes that activate diverse carboxylic acid substrates to produce acyl-CoAs. We hypothesized that Sl-AACS1 uses medium chain fatty acids as substrates, because ablation of *Sl-AACS1* eliminated acylsugars with medium acyl chains. To characterize the in vitro activity of Sl-AACS1, we purified recombinant His-tagged proteins from *Escherichia coli*. Enzyme assays were performed by supplying fatty acid substrates with even carbon numbers from C2 through C18 (*Figure 4B*). The results showed that Sl-AACS1 utilized fatty acid substrates with lengths ranging from C6 to C12, including those with a terminal branched carbon (iC10:0) or an unsaturated bond (*trans*-2-decenoic acid, C10:1) (*Figure 4*, B and C). However, no activity was observed with the 3-hydroxylated C12 and C14 fatty acids as substrates (*Figure 4B*). These results support our hypothesis that Sl-AACS1 produces medium chain acyl-CoAs, which are in vivo substrates for acylsugar biosynthesis.

To test whether *Sl-AACS1* and *Sl-AECH1* can produce medium chain acyl-CoAs in planta, we transiently expressed these genes in *Nicotiana benthamiana* leaves using *Agrobacterium*-mediated

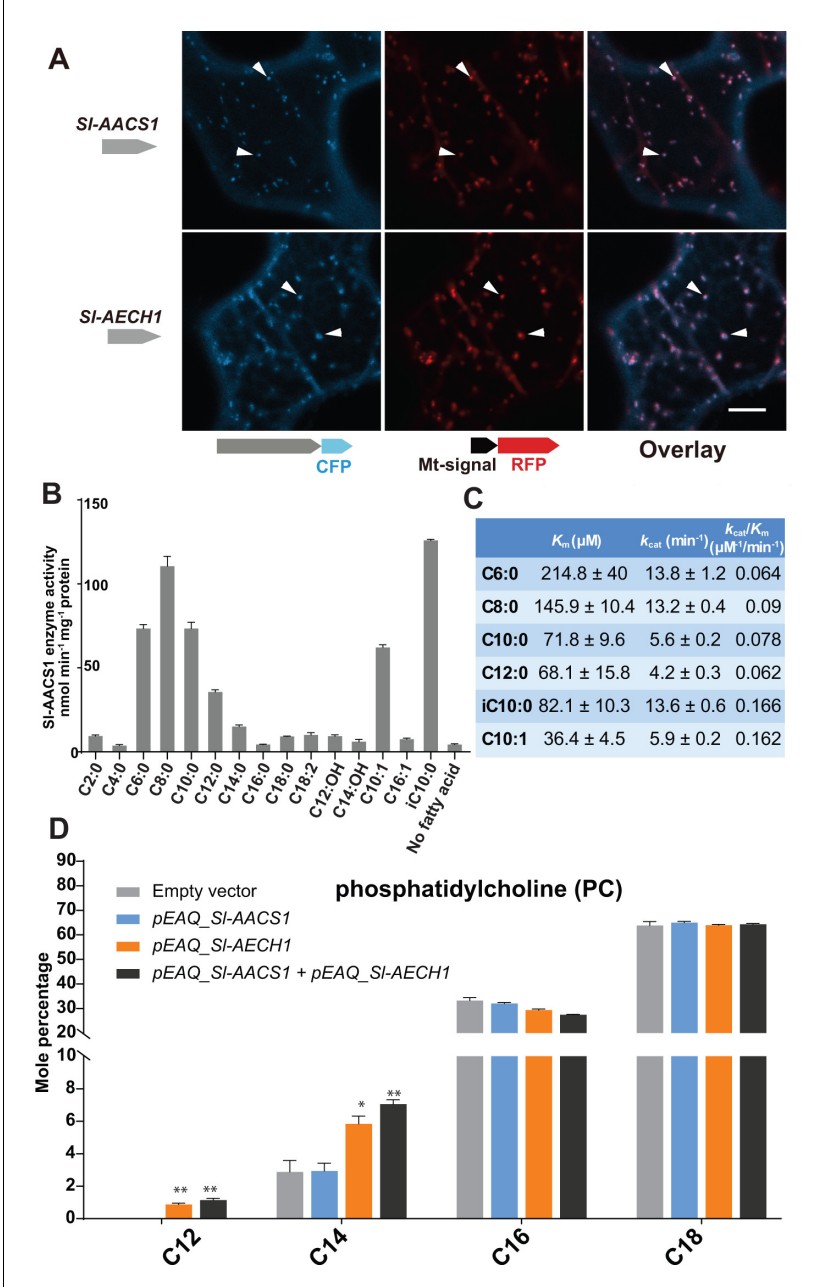

**Figure 4.** Functional analysis of Sl-AACS1 and Sl-AECH1 in *N. benthamiana* and recombinant Sl-AACS1 enzyme analysis. (**A**) Confocal images of co-expression analysis in tobacco leaf epidermal cells using C-terminal CFP-tagged either Sl-AACS1 or Sl-AECH1 and the mitochondrial marker MT-RFP. Arrowheads point to mitochondria that are indicated by MT-RFP fluorescent signals. Scale bar equals 10 μm. *Figure 4—figure supplement 1B* describes that the expressed YFP-recombinant proteins were not co-localized with the peroxisomal marker RFP-PTS (**B**) Aliphatic fatty acids of different chain lengths were used as the substrates to test Sl-AACS1 acyl-CoA synthetase activity. Mean amount of acyl-CoAs generated (nmol min$^{-1}$ mg$^{-1}$ proteins) was used to represent enzyme activities. The results are from three measurements ± SEM. *Figure 4—source data 1* includes values for the measurements. (**C**) Enzyme activity of Sl-AACS1 for six fatty acid substrates. (**D**) Identification of membrane lipid phosphatidylcholine (PC), which contains medium acyl chains, following transient expression of *Sl-AECH1* in *N. benthamiana* leaves. The results from expressing *Sl-AACS1* and co-expressing both *Sl-AECH1* and *Sl-AACS1* are also shown. Mole percentage (Mol %) of the acyl chains from membrane lipids with carbon number 12, 14, 16, and 18 are shown for three biological replicates ± SEM. *p<0.05, **p<0.01. Welch two-sample *t* test was performed comparing with the empty vector control. *Figure 4—source data 2* includes values for the lipid analysis. Acyl groups of the same chain lengths with saturated and unsaturated bonds were combined in the

*Figure 4 continued on next page*

*Figure 4 continued*

calculation. *Figure 4—figure supplement 2* shows that the putative *Sl-AECH1* orthologs from *S. pennellii* and *S. quitoense* generated medium chain lipids in the infiltrated leaves.

The online version of this article includes the following source data and figure supplement(s) for figure 4:

**Source data 1.** Data used to make *Figure 4B*.
**Source data 2.** Data used to make *Figure 4D* and *Figure 4—figure supplement 1C*.
**Figure supplement 1.** Characterization of cluster genes using leaf transient expression: protein subcellular targeting and impacts on lipid metabolism.
**Figure supplement 2.** The closest homologs of *Sl-AECH1* from other *Solanum* species generate medium chain lipids when transiently expressed in *N. benthamiana*.

infiltration (*Sainsbury et al., 2009*). It is challenging to directly measure plant acyl-CoAs, due to their low concentration and separate organellar pools. We used an alternative approach and characterized membrane lipids, which are produced from acyl-CoA intermediates. We took advantage of the observation that *N. benthamiana* membrane lipids do not accumulate detectable acyl chains of 12 carbons or shorter. *N. benthamiana* leaves were infiltrated with constructs containing *Sl-AACS1* or *Sl-AECH1* individually, or together (*Figure 4D*). In contrast to the empty vector control, infiltration of *Sl-AECH1* led to detectable levels of C12 acyl chains in the leaf membrane lipid phosphatidylcholine (PC) (*Figure 4D*). We also observed increased C14 acyl chains in PC, phosphatidylglycerol (PG), sulfoquinovosyl diacylglycerol (SQDG), and digalactosyldiacylglycerol (DGDG) in *Sl-AECH1* infiltrated plants (*Figure 4D* and *Figure 4—figure supplement 1C*). These results suggest that *Sl-AECH1* participates in generation of medium chain acyl-CoAs in planta, which are channeled into lipid biosynthesis. No medium chain acylsugars were detected, presumably due to the lack of core acylsugar biosynthetic machinery in *N. benthamiana* mesophyll cells.

We asked whether the closest known homologs of *Sl-AECH1* from *Solanum* species can generate medium chain lipids when transiently expressed in *N. benthamiana*. Two SQDGs with C12 chains were monitored by LC/MS as peaks diagnostic of lipids containing medium chain fatty acids (*Figure 4—figure supplement 2, A and B*). The results showed that only the putative *Sl-AECH1* orthologs *Sopen07g023250* (*Sp-AECH1*) and *Sq_c37194* (*Sq-AECH1*) – from *S. pennellii* and *S. quitoense* respectively – generated medium chain lipids in the infiltrated leaves (*Figure 4—figure supplement 2C*). This confirms that not all ECHs can produce medium chain lipids and suggests that the function of *Sl-AECH1* evolved recently, presumably as a result of neofunctionalization after gene duplication (*Figure 4—figure supplement 2C*).

## *AACS1* and *AECH1* are evolutionarily conserved in the *Solanum*

Medium chain acylsugars were documented in *Solanum* species besides cultivated tomato, including *S. pennellii* (*Leong et al., 2019*), *S. nigrum* (*Moghe et al., 2017*), as well as the more distantly related *S. quitoense* (*Leong et al., 2020*; *Hurney, 2018*) and *S. lanceolatum* (*Herrera-Salgado et al., 2005*). We hypothesized that evolution of *AACS1* and *AECH1* contributed to medium chain acylsugar biosynthesis in *Solanum*. As a test, we analyzed the genomes of *Solanum* species other than cultivated tomato. Indeed, the acylsugar related synteny containing ACS and ECH was found in both *S. pennellii* and *S. melongena* (eggplant), suggesting that the cluster assembly evolved before divergence of the tomato and eggplant lineage (*Figure 5A*).

We applied gene expression and genetic approaches to test the in vivo functions of ACS and ECH in selected *Solanum* species. To explore the expression pattern of *S. pennellii* ACS and ECH cluster genes, we performed RNA-seq analysis on trichomes and shaved stems to identify acylsugar biosynthetic candidates (*Supplementary file 2*). The expression pattern of *S. pennellii* cluster genes is strikingly similar to *S. lycopersicum*: one ECH and two ACS genes are highly enriched in trichomes, including the orthologs of *Sl-AACS1* and *Sl-AECH1*. *Sp-AACS1* function (*Sopen07g023200*) was first tested by asking whether it can reverse the cultivated tomato *sl-aacs1* mutant acylsugar phenotype. Indeed, *Sp-AACS1* restored C12 containing acylsugars in the stably transformed *sl-aacs1* plants (*Figure 5—figure supplement 1*). To directly test *Sp-AACS1* and *Sp-AECH1* function, we used CRISPR-Cas9 to make single mutants in *S. pennellii* LA0716. No medium chain acylsugars were detected in T0 generation mutants with edits for each gene (*Figure 5B* and *Figure 5—figure supplement 2, A and C*). Similar to the ACS-annotated *Solyc07g043660* cultivated tomato mutant (*Figure 3—figure*

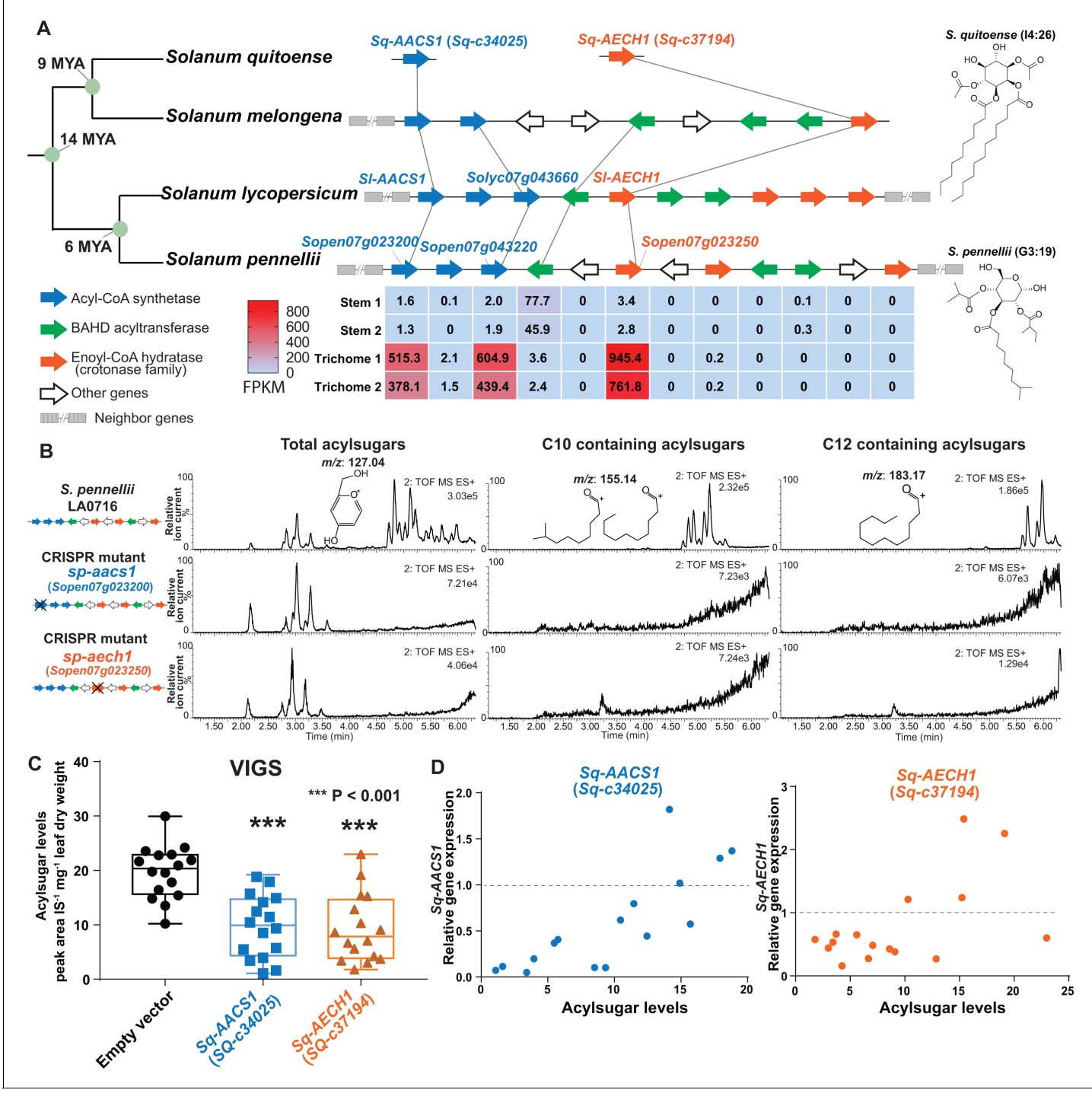

**Figure 5.** *AACS1* and *AECH1* are evolutionarily conserved in *Solanum* plants. (**A**) A conserved syntenic genomic region containing *AACS1* and *AECH1* was found in three selected *Solanum* species. Nodes representing estimated dates since the last common ancestors (*Särkinen et al., 2013*) shown on the left. The closest homologs of *AACS1* and *AECH1* in *Solanum quitoense* are shown without genomic context because the genes were identified from RNA-seq and genome sequences are not available. The lines connect genes representing putative orthologs across the four species. The trichome/stem RNA-seq data of two biological *S. pennellii* replicates are summarized (*Supplementary file 2*) for genes in the syntenic region. The red-blue color gradient provides a visual marker to rank the expression levels in FPKM. Structures of representative medium chain acylsugars from *S. quitoense* (acylinositol, I4:26) (*Hurney, 2018*) and *S. pennellii* (acylglucose, G3:19) (*Leong et al., 2019*) are on the right. *Figure 5—figure supplement 1* shows that stable *Sp-AACS1* transformation of the M82 CRISPR mutant *sl-aacs1* restores C12 containing acylsugars (**B**) CRISPR/Cas9-mediated gene knockout of *Sp-AACS1* or *Sp-AECH1* in *S. pennellii* produce no detectable medium chain containing acylsugars. The ESI⁺ mode LC/MS extracted ion chromatograms of trichome metabolites are shown for each mutant. The *m/z* 127.01 (left panel) corresponds to the glucopyranose ring fragment that

*Figure 5 continued on next page*

*Figure 5 continued*

both acylsucroses and acylglucoses generate under high collision energy positive-ion mode. The *m/z* 155.14 (center panel) and 183.17 (right panel) correspond to the acylium ions from acylsugars with chain length of C10 and C12, respectively. *Figure 5—figure supplement 2A–C* describes the design of the gRNAs and the detailed information of gene edits. (C) Silencing *Sq-AACS1* (*Sq-c34025*) or *Sq-AECH1* (*Sq-c37194*) in *S. quitoense* using VIGS leads to reduction of total acylsugars. The peak area/internal standard (IS) normalized by leaf dry weight was shown from sixteen plants ± SEM. \*\*\*p<0.001, Welch two-sample *t* test. *Figure 5—figure supplement 2E and F* describes the VIGS experimental design and the representative LC/MS extracted ion chromatograms of *S. quitoense* major acylsugars. (D) Reduced gene expression of *Sq-AACS1* or *Sq-AECH1* correlates with decreased acylsugar levels in *S. quitoense*. The qRT-PCR gene expression data are plotted with acylsugar levels of the same leaf as described in *Figure 5—figure supplement 2E*. *Figure 5—source data 1* includes raw data for the *S. quitoense* VIGS experiments.

The online version of this article includes the following source data and figure supplement(s) for figure 5:

**Source data 1.** Data used to make *Figure 5C and D*.
**Figure supplement 1.** Stable *Sp-AACS1* transformation of the M82 CRISPR mutant *sl-aacs1* restores C12 containing acylsugars.
**Figure supplement 2.** Functional analysis of *AACS1* and *AECH1* in *S. pennellii* and *S. quitoense* via CRISPR-Cas9 system and VIGS, respectively.

---

*supplement 1D*), deletion of *S. pennellii* ortholog *Sopen07g023220* has no observed effects on *S. pennellii* trichome acylsugars (*Figure 5—figure supplement 2D*).

The medium chain acylsugar producer *S. quitoense* (*Hurney, 2018*) was used for *AACS1* and *AECH1* functional analysis because of its phylogenetic distance from the tomato clade - it is in the *Solanum* Leptostemonum clade (including eggplant) - and the fact that it produces medium chain acylsugars. We found trichome-enriched putative orthologs of *AACS1* and *AECH1* in the transcriptome dataset of *S. quitoense* (*Moghe et al., 2017*), and tested their in vivo function through virus-induced gene silencing (VIGS) (*Figure 5—figure supplement 2E*). Silencing either gene led to decreased total acylsugars (*Figure 5C* and *Figure 5—figure supplement 2F*), which correlated with the degree of expression reduction in each sample (*Figure 5D*). These results are consistent with the hypothesis that *Sq-AACS1* and *Sq-AECH1* are involved in medium chain acylsugar biosynthesis, because all acylsugars in *S. quitoense* carry two medium chains (*Leong et al., 2020*; *Hurney, 2018*). The importance of *AACS1* and *AECH1* in medium chain acylsugar biosynthesis in distinct *Solanum* clades inspired us to explore the evolutionary origins of the gene cluster.

## Evolution of the gene cluster correlates with the distribution of medium chain acylsugars across Solanaceae

We sought to understand how the acylsugar gene cluster evolved and whether it correlates with the distribution of medium chain acylsugars across the Solanaceae family. Taking advantage of the available genome sequences of 13 species from Solanaceae and sister families, we analyzed the regions that are syntenic with the tomato acylsugar gene cluster (*Figure 6—figure supplement 1*). This synteny was found in all these plants, including the most distantly related species analyzed, *Coffea canephora* (coffee, Rubiaceae) (*Figure 6—figure supplement 1*). BAHD acyltransferases were the only genes observed in the syntenic regions both inside and outside the Solanaceae, in contrast to ECH and ACS, which are restricted to the family (*Figure 6A* and *Figure 6—figure supplement 1*). Within the syntenic regions of the species analyzed, ECH homologs, including pseudogenes, are present in all Solanaceae except for *Capsicum* species, while ACS is more phylogenetically restricted, being found only in *Nicotiana* and *Solanum* (*Figure 6A* and *Figure 6—figure supplement 1*).

We then performed phylogenetic analysis to reconstruct the evolutionary history of the ACS, ECH, and BAHD acyltransferase genes in the syntenic region (*Figure 6*). This analysis revealed a model for the temporal order of emergence for the three types of genes, leading to their presence in the syntenic regions in extant Solanaceae plants (*Figure 6B*). We propose that the BAHD acyltransferase was the first of three genes that emerged before the divergence between Solanaceae and Rubiaceae, and was likely lost in Convolvulaceae. This hypothesis is based on the discovery of a BAHD acyltransferase pseudogene in the syntenic region of *C. canephora* (*Figure 6A* and *Figure 6—figure supplement 1*), which is one of the closest *Coffea* sequences sister to the ASAT clade (*Figure 6—figure supplement 2* and *Figure 6—figure supplement 5*). In our model, ECH was likely inserted into the syntenic region before the Solanaceae-specific whole genome duplication (WGD) event (*Figure 6B* and *Figure 6—figure supplement 3*).

We propose that ACS was inserted into the synteny through segmental duplication (*Bailey et al., 2002*; *Figure 6—figure supplement 4*). However, whether ACS insertion happened before or after

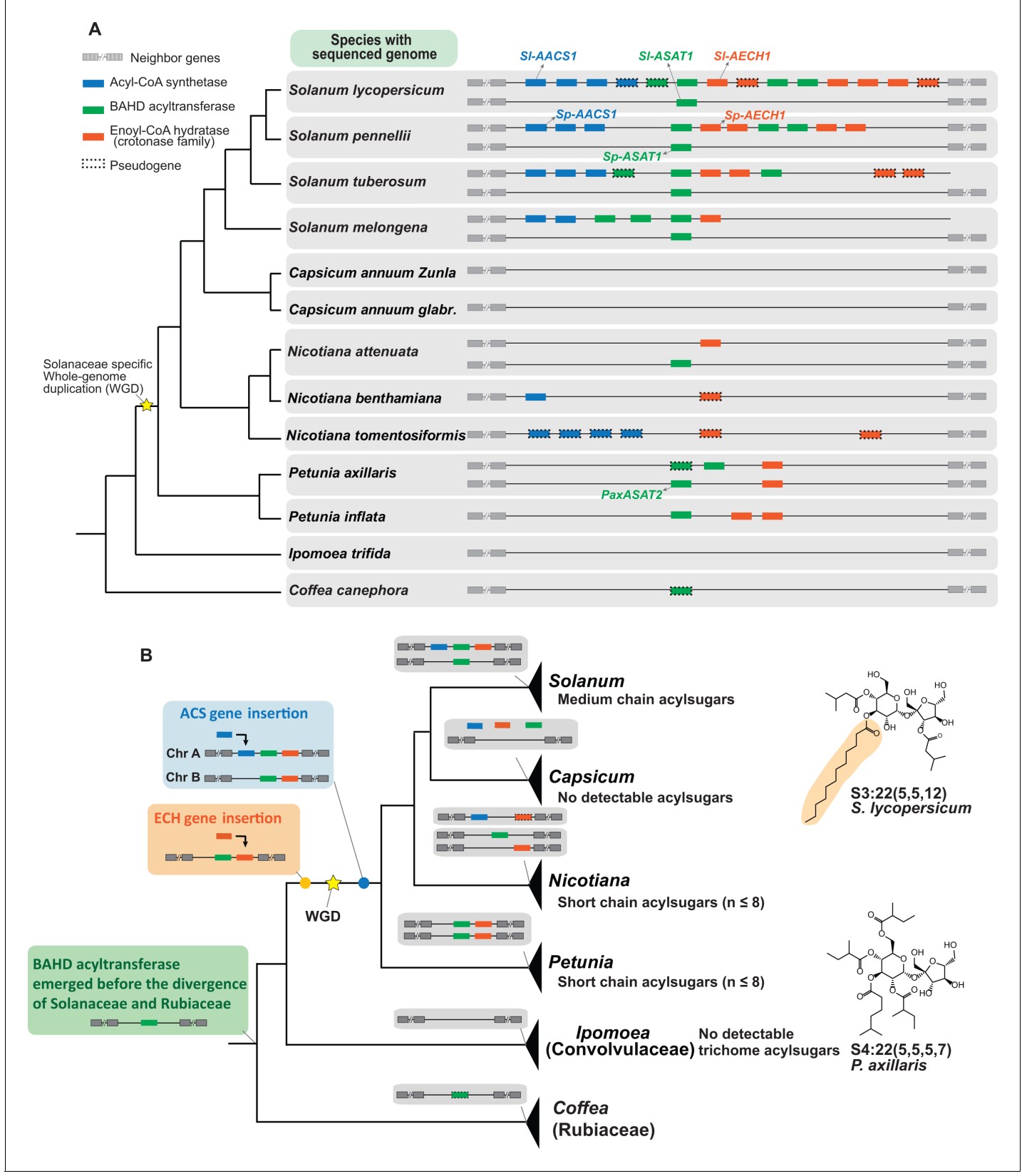

**Figure 6.** Evolution of the acylsugar gene cluster is associated with acylsugar acyl chain diversity across the Solanaceae family. (**A**) The acylsugar gene cluster syntenic regions of 11 Solanaceae species and two outgroup species *Ipomea trifida* (Convolvulaceae) and *Coffea canephora* (Rubiaceae). This is a simplified version adapted from *Figure 6—figure supplement 1*. Only genes from the three families – ACS (blue), BAHD acyltransferase (green), and

*Figure 6 continued*

ECH (orange) – are shown. For information about the syntenic region size in each species refer to *Figure 6—figure supplement 1* and *Supplementary file 4*. (B) The evolutionary history of the acylsugar gene cluster and its relation to the acylsugar phenotypic diversity. The evolution of BAHD acyltransferase genes is inferred based on *Figure 6—figure supplement 2* and *Figure 6—figure supplement 5*. ECH genes based on *Figure 6—figure supplement 3*. ACS genes based on *Figure 6—figure supplement 4* and *Figure 6—figure supplement 6*. The temporal order for the emergence for the three types of genes are shown in the colored boxes on the left: green box (BAHD acyltransferase), orange box (ECH), blue box (ACS). The yellow star represents the Solanaceae-specific WGD. Structures of representative short and medium chain acylsugars were shown on the right. *Figure 6—figure supplement 8* describes distribution of acylsugar acyl chains with different lengths in species across the Solanaceae family.

The online version of this article includes the following figure supplement(s) for figure 6:

**Figure supplement 1.** Syntenic regions containing the acylsugar gene cluster.
**Figure supplement 2.** Analysis of the evolutionary history of the BAHD acyltransferases in the syntenic regions in different Solanaceae species.
**Figure supplement 3.** Analysis of the evolutionary history of the ECH genes in the syntenic regions in different Solanaceae species.
**Figure supplement 4.** Analysis of the evolutionary history of the ACS genes in the syntenic regions in different Solanaceae species.
**Figure supplement 5.** Phylogenetic analysis of the BAHD acyltransferase.
**Figure supplement 6.** Additional evolutionary analysis of ACS genes in the syntenic regions to understand when the segmental duplication event happened.
**Figure supplement 7.** Ancestral trait state reconstruction analysis.
**Figure supplement 8.** Phylogenetic distribution of acylsugar acyl chains with different lengths across the Solanaceae family.

the Solanaceae-specific WGD event cannot be resolved by the phylogenetic analysis (*Figure 6—figure supplement 4*). If the insertion happened before WGD, one ACS gene loss on chromosome 12 in *Solanum* – as well as two independent gene losses on chromosomes 7 and 12 in both *Petunia* (*Figure 6—figure supplement 4*) and in *Salpiglossis sinuata* (*Figure 6—figure supplement 6*) – should have happened. However, if the insertion happened after WGD, then only one gene loss in *Petunia* and *Salpiglossis* needs to be invoked (*Figure 6—figure supplement 6*). The latter scenario is more likely based on the principle of parsimony.

Our ancestral state reconstruction inference supports the notion that the medium chain acylsugars co-emerged with the ACS/ECH genes in the syntenic regions in the common ancestor of *Solanum* (*Figure 6—figure supplement 7*). This leads us to propose that the emergence of both ACS and ECH genes in the synteny was a prerequisite for the rise of medium chain acylsugars in Solanaceae species (*Figure 6B*). Consistent with the hypothesis, only short chain acylsugars were observed in *Petunia* (*Liu et al., 2017*; *Figure 6—figure supplement 8*), which correlates with the absence of ACS homolog (*Figure 6B*). In contrast, medium chain acylsugars were detected throughout *Solanum* (*Figure 6—figure supplement 8*), supported by the observation that both ACS and ECH homologs are present in extant *Solanum* species (*Figure 6B*). Interestingly, although *Nicotiana* species collectively have both ACS and ECH genes (*Figure 6B*), they do not produce medium chain acylsugars (*Figure 6—figure supplement 8*) presumably due to gene losses. For example, the ECH homologs are pseudogenes in *N. benthamiana* and *N. tomentosiformis* (*Figure 6A*). These results show that the presence of both functional ACS and ECH genes are associated with the accumulation of medium chain acylsugars, supporting our hypothesis above.

Although no medium chain acylsugars were detected in *Nicotiana* species examined, the ACS and ECH genes may have been present in the syntenic region prior to divergence of *Solanum* and *Nicotiana*. This suggests that one or more species that diverged from the common ancestor of *Solanum* and *Nicotiana* could have medium chain acylsugars. To test this hypothesis, we extended the phenotypic analysis to six such Solanaceae genera (*Figure 6—figure supplement 8*). Indeed, we found that species in *Jaltomata*, *Physalis*, *Iochroma*, *Atropa*, and *Hyoscyamus*, which diverged from the common ancestor with *Nicotiana* but before *Solanum*, have medium chain acylsugars (*Figure 6—figure supplement 8*).

## Discussion

This study identified a *S. lycopersicum* chromosome 7 synteny of ACS, ECH, and BAHD acyltransferase genes including two involved in medium chain acylsugar biosynthesis. The discovery of this locus was prompted by our observation of increased C10 containing acylsugars in tomato recombinant lines carrying this region from the wild tomato *S. pennellii* chromosome 7. In vitro biochemistry

revealed that Sl-AACS1 produces acyl-CoAs using C6-C12 fatty acids as substrates. The function of *AACS1* and *AECH1* in cultivated and wild tomato medium chain acylsugar biosynthesis was confirmed by genome editing, and extended to the phylogenetically distant *S. quitoense* using VIGS. The trichome tip cell-specific expression of these genes is similar to that of previously characterized acylsugar pathway genes (*Fan et al., 2019*).

There are increasing examples of plant specialized metabolic innovation evolving from gene duplication and neofunctionalization of primary metabolic enzymes (*Maeda, 2019*; *Milo and Last, 2012*; *Moghe and Last, 2015*; *Zi et al., 2014*). Recruitment of *Sl-AACS1* and *Sl-AECH1* from fatty acid metabolism provides new examples of 'hijacking' primary metabolic genes into acylsugar biosynthesis, in addition to an isopropylmalate synthase (*Sl-IPMS3*) and an invertase (*Sp-ASFF1*) (*Leong et al., 2019*; *Ning et al., 2015*). We hypothesize that both AACS1 and AECH1 participate in generation of medium chain acyl-CoAs, the acyl donor substrates for ASAT-catalyzed acylsugar biosynthesis. Indeed, Sl-AACS1 exhibits in vitro function consistent with this hypothesis, efficiently utilizing medium chain fatty acids as substrates to synthesize acyl-CoAs. Strikingly, Sl-AECH1 perturbs membrane lipid composition when transiently expressed in *N. benthamiana* leaves, generating unusual C12-chain membrane lipids.

These results suggest that evolution of trichome tip cell-specific gene expression potentiated the co-option of *AACS1* and *AECH1* in medium chain acylsugar biosynthesis. Analogous to trichomes producing medium chain acylsugars, seeds of phylogenetically diverse plants accumulate medium chain fatty acid storage lipids (*Ohlrogge et al., 2018*). In contrast, fatty acids with unusual structures, including those of medium chain lengths, are rarely found in membrane lipids, presumably because these would perturb membrane bilayer structure and function (*Millar et al., 2000*). For example, seed embryo-specific expression of three neofunctionalized enzyme variant genes in *Cuphea* species – an acyl-ACP thioesterase (*Dehesh et al., 1996*), a 3-ketoacyl-ACP synthase (*Dehesh et al., 1998*), and a diacylglycerol acyltransferase (*Iskandarov et al., 2017*) – lead to production of medium chain seed storage lipids (*Voelker and Kinney, 2001*). Trichome tip cell restricted expression of *AACS1* and *AECH1* represents an analogous example of diverting neofunctionalized fatty acid enzymes from general metabolism into cell-specific specialized metabolism. It is notable that we obtained evidence that Sl-AACS1 and Sl-AECH1 are targeted to mitochondria. Because the other characterized acylsugar biosynthetic enzymes – ASATs and Sl-IPMS3 – appear to be cytoplasmic, these results suggest that medium chain acylsugar substrates are intracellularly transported within the trichome tip cell. It is worth noting that Sl-AACS1 seems to show higher activity with C8 fatty acid than with C10 or C12 (*Figure 4*, B and C), while no C8 containing acylsugars were described in tomato trichomes (*Ghosh et al., 2014*). This suggests that C8 fatty acids are not synthesized in trichomes.

Beyond employing functional approaches, this study demonstrates the value of a combined comparative genomic and metabolomic analysis in reconstructing the evolutionary history of a gene cluster: in this case over tens of millions of years. We propose that the acylsugar gene cluster started with a 'founder' BAHD acyltransferase gene, followed by sequential insertion of ECH and ACS (*Figure 6B*). This de novo assembly process is analogous to evolution of the antimicrobial triterpenoid avenacin cluster in oat (*Qi et al., 2006*; *Qi et al., 2004*). There are two noteworthy features of our approach. First, reconstructing acylsugar gene cluster evolution in a phylogenetic context allows us to deduce cluster composition in extant species (*Figure 6B*). Second, it links cluster genotype with acylsugar phenotype and allows inference of acylsugar diversity across the Solanaceae (*Figure 6* and *Figure 6—figure supplement 8*).

The current architecture of the Solanaceae acylsugar synteny merely represents a snapshot of a genomic neighborhood that is dynamic, which echoes a recent study of triterpene biosynthetic gene clusters in the Brassicaceae (*Liu et al., 2019*; *Peters, 2020*). De novo assembly of the gene cluster was accompanied by gene duplication, transposition, pseudogenization, and deletion in different genera. In the case of non-acylsugar producer *Capsicum*, although phylogenetic analysis revealed putative *Sl-AACS1* and *Sl-AECH1* orthologous genes, they are not harbored in the syntenic region, probably due to translocation or assembly quality issues (*Figure 6A* and *Figure 6—figure supplement 1*). In *Nicotiana*, the ECH genes became pseudogenized (*Figure 6B*), which is associated with lack of detectable plant medium chain acylsugars (*Figure 6—figure supplement 8*). In tomatoes, the trichome expressed *Solyc07g043660* derives from a recent tandem duplication (*Figure 6—figure supplement 4*), yet its deletion has no effect on trichome acylsugars (*Figure 3—figure supplement 1D*). A parsimonious explanation is that *Solyc07g043660* is experiencing functional

divergence, which may eventually lead to pseudogenization as observed for other genes in the syntenic region.

In this study, we identified an acylsugar SMGC in the context of a multiple chromosome syntenic region. This synteny resulted from WGD, and the acylsugar-related genes are co-expressed, and involved in the same metabolic pathway. This resembles the tomato steroidal alkaloid gene cluster consisting of eight genes that are dispersed into two syntenic chromosome regions (*Itkin et al., 2013*). In fact, this tomato alkaloid SMGC is located next to the acylsugar cluster (*Figure 3—figure supplement 2*), which is reminiscent of two physically adjacent SMGCs in the fungus *Aspergillus* (*Wiemann et al., 2013*). Tomato steroidal alkaloids and acylsugars both serve defensive roles in plants, but are biosynthetically and structurally distinct and are stored in different tissues. This raises intriguing questions. Did the separation of acylsugar and alkaloid SMGCs into two chromosomes occur contemporaneously and by the same mechanism? Did this colocalization confer selective advantage through additive or synergistic effects of multiple classes of defensive metabolites? Answering these questions requires continued mining and functional validation of metabolic gene clusters across broader plant species and analysis of impacts of clustering in evolutionary and ecological contexts.

## Materials and methods

**Key resources table**

| Reagent type (species) or resource | Designation | Source or reference | Identifiers | Additional information |
|---|---|---|---|---|
| Gene (*Solanum lycopersicum* M82) | *Sl-AACS1* | This paper | GeneBank: MT078737 | Characterized and named in the results |
| Gene (*Solanum lycopersicum* M82) | *Sl-AECH1* | This paper | GeneBank: MT078736 | Characterized and named in the results |
| Gene (*Solanum pennellii* LA0716) | *Sp-AACS1* | This paper | GeneBank: MT078735 | Characterized and named in the results |
| Gene (*Solanum pennellii* LA0716) | *Sp-AECH1* | This paper | GeneBank: MT078734 | Characterized and named in the results |
| Gene (*Solanum quitoense*) | *Sq-AACS1* | This paper | GeneBank: MT078732 | Characterized and named in the results |
| Gene (*Solanum quitoense*) | *Sq-AECH1* | This paper | GeneBank: MT078731 | Characterized and named in the results |
| Gene (*Solanum quitoense*) | *Sq_c35719* | This paper | GeneBank: MT078733 | Characterized and named in the results |
| Software, algorithm | Trimmomatic | http://www.usadellab.org/cms/index.php?page=trimmomatic | RRID:SCR_011848 | |
| Software, algorithm | TopHat | http://ccb.jhu.edu/software/tophat/index.shtml | RRID:SCR_013035 | |
| Software, algorithm | Cufflinks | http://cole-trapnell-lab.github.io/cufflinks/cuffmerge/ | RRID:SCR_014597 | |
| Software, algorithm | MCScanX-transposed | http://chibba.pgml.uga.edu/mcscan2/transposed/ | | |
| Software, algorithm | RAxML | https://github.com/stamatak/standard-RAxML | RRID:SCR_006086 | |

*Continued on next page*

*Continued*

| Reagent type (species) or resource | Designation | Source or reference | Identifiers | Additional information |
|---|---|---|---|---|
| Software, algorithm | Mesquite | https://www.mesquiteproject.org/ | RRID:SCR_017994 | |

## Plant materials and trichome metabolite extraction

The seeds of cultivated tomato *Solanum lycopersicum* M82 were obtained from the C.M. Rick Tomato Genetic Resource Center (https://tgrc.ucdavis.edu/), RRID:SCR_014954. Tomato introgression lines (ILs) and tomato backcross inbred lines (BILs) were from Dr. Dani Zamir (Hebrew University of Jerusalem). The tomato seeds were treated with ½ strength bleach for 30 min and washed with de-ionized water three or more times before placing on wet filter paper in a Petri dish. After germination, the seedlings were transferred to peat-based propagation mix (Sun Gro Horticulture) and transferred to a growth chamber for two or three weeks under 16 hr photoperiod, 28°C day and 22°C night temperatures, 50% relative humidity, and 300 µmol m$^{-2}$ s$^{-1}$ photosynthetic photon flux density. The youngest fully developed leaf was submerged in 1 mL extraction solution in a 1.5 mL screw cap tube and agitated gently for 2 min. The extraction solution contains acetonitrile/isopropanol/water (3:3:2) with 0.1% formic acid and 10 µM propyl-4-hydroxybenzoate as internal standard. The interactive protocol of acylsugar extraction is available in Protocols.io at http://dx.doi.org/10.17504/protocols.io.xj2fkqe.

## DNA construct assembly and tomato transformation

Assembly of the CRISPR-Cas9 constructs was as described (*Brooks et al., 2014*; *Leong et al., 2019*). Two guide RNAs (gRNAs) were designed targeting one or two exons of each gene to be knocked out by the CRISPR-Cas9 system. The gRNAs were obtained from gene blocks (gBlocks) synthesized by IDT (Integrated DNA Technologies, location) (*Supplementary file 3*). For each CRISPR construct, two gBlocks and four plasmids from Addgene, pICH47742::2 × 35 S-5′UTR-hCas9 (STOP)-NOST (Addgene no. 49771), pICH41780 (Addgene no. 48019), pAGM4723 (Addgene no. 48015), pICSL11024 (Addgene no. 51144), were mixed for DNA assembly using the Golden Gate assembly kit (NEB).

For in planta tissue specific reporter gene analysis, 1.8 kb upstream of the annotated translational start site of *Sl-AACS1* and *Sl-AECH* were amplified using the primer pairs SlAACS1-pro_F/R and SlAECH1-pro_F/R (*Supplementary file 3*). The amplicon was inserted into the entry vector pENTR/D-TOPO, followed by cloning into the GATEWAY vector pKGWFS7. For ectopically expressing *Sp-AACS1* in the cultivated tomato CRISPR mutant *sl-aacs1*, *Sp-AACS1* gene including 1.8 kb upstream of the translational start site of *Sp-AACS1* was amplified using the primer pair SpAACS1-pro-gene_F/R (*Supplementary file 3*). The amplicon was inserted into the entry vector pENTR/D-TOPO, followed by cloning into the GATEWAY vector pK7WG.

The plant transformation of *S. lycopersicum* M82 and *S. pennellii* LA0716 was performed using the *Agrobacterium tumefaciens* strain AGL0 following published protocols (*Leong et al., 2019*; *McCormick, 1997*). The primers used for genotyping the *S. lycopersicum* M82 transgenic plants harboring pK7WG or pKGWFS7 construct are listed in *Supplementary file 3*. For genotyping the *S. lycopersicum* M82 CRISPR mutants in the T1 generation, the sequencing primers listed in *Supplementary file 3* were used to amplify the genomic regions harboring the gRNAs and the resultant PCR products were sent for Sanger sequencing. For genotyping the *S. pennellii* LA0716 CRISPR mutants in the T0 generation, the sequencing primers listed in *Supplementary file 3* were used to amplify the genomic regions harboring the gRNAs. The resulting PCR products were cloned into the pGEM-T easy vector (Promega) and transformed into *E. coli*. Plasmids from at least six individual *E. coli* colonies containing the amplified products were extracted and verified by Sanger sequencing.

## Protein subcellular targeting in tobacco mesophyll cells

For protein subcellular targeting analysis, the open reading frame (ORF) of *Sl-AACS1*, *Sl-AECH1*, and *Solyc07g043660* were amplified using the primers listed in *Supplementary file 3*. These amplicons were inserted into pENTR/D-TOPO respectively, followed by subcloning into the GATEWAY

vectors pEarleyGate102 (no. CD3-684) and pEarleyGate104 (no. CD3-686), which were obtained from Arabidopsis Biological Resource Center (ABRC). For the pEarleyGate102 constructs, the CFP was fused to the C-terminal of the tested proteins. For the pEarleyGate104 constructs, the YFP was fused to the protein N-terminus. Transient expressing the tested proteins was performed following an established protocol (*Batoko et al., 2000*) with minor modifications. In brief, cultures of *A. tumefaciens* (strain GV3101) harboring the expression vectors were washed and resuspended with the infiltration buffer (20 mM acetosyringone, 50 mM MES pH 5.7, 0.5% glucose [w/v] and 2 mM $Na_3PO_4$) to reach $OD_{600nm}$ = 0.05. Four-week-old tobacco (*Nicotiana tabacum* cv. Petit Havana) plants grew in 21°C and 8 hr short-day conditions were infiltrated, and then maintained in the same growth condition for three days before being sampled for imaging. The GV3101 cultures containing the mitochondria marker MT-RFP (*Nelson et al., 2007*) were co-infiltrated to provide the control signals for mitochondrial targeting. In separate experiments, the GV3101 cultures containing the peroxisome marker RFP-PTS (*Nelson et al., 2007*) were co-infiltrated to provide the control signals for peroxisomal targeting.

## Confocal microscopy

A Nikon A1Rsi laser scanning confocal microscope and Nikon NIS-Elements Advanced Research software were used for image acquisition and handling. For visualizing GFP fluorescence in trichomes of the tomato transformants, the excitation wavelength at 488 nm and a 505- to 525 nm emission filter were used for the acquisition. For visualizing signals of fluorescence proteins in the tobacco mesophyll cells, CFP, YFP and RFP, respectively, were detected by excitation lasers of 443 nm, 513 nm, 561 nm and emission filters of 467–502 nm, 521–554 nm, 580–630 nm.

## *N. benthamiana* transient gene expression and membrane lipid analysis

For *N. benthamiana* transient expression of *Sl-AACS1*, *Sl-AECH1*, and homologs of *AECH1*, the ORFs of these genes were amplified using primers listed in *Supplementary file 3*, followed by subcloning into pEAQ-HT vector using the Gibson assembly kit (NEB). Linearization of pEAQ-HT vector was performed by XhoI and AgeI restriction enzyme double digestion. *A. tumefaciens* (strain GV3101) harboring the pEAQ-HT constructs were grown in LB medium containing 50 μg/mL kanamycin, 30 μg/mL gentamicin, and 50 μg/ml rifampicin at 30 °C. The cells were collected by centrifugation at 5000 g for 5 min and washed once with the resuspension buffer (10 mM MES buffer pH 5.6, 10 mM $MgCl_2$, 150 μM acetosyringone). The cell pellet was resuspended in the resuspension buffer to reach $OD_{600nm}$ = 0.5 for each strain and was incubated at room temperature for 1 hr prior to infiltration. Leaves of 4 to 5 week-old *N. benthamiana* grown under 16 hr photoperiod were used for infiltration. Five days post infiltration, the infiltrated leaves were harvested, ground in liquid nitrogen, and stored at −80 °C for later analysis.

The membrane lipid analysis was performed as previously described (*Wang and Benning, 2011*). In brief, the *N. benthamiana* leaf polar lipids were extracted in the organic solvent containing methanol, chloroform, and formic acid (20:10:1, v/v/v), separated by thin layer chromatography (TLC), converted to fatty acyl methylesters (FAMEs), and analyzed by gas-liquid chromatography (GLC) coupled with flame ionization. The TLC plates (TLC Silica gel 60, EMD Chemical) were activated by ammonium sulfate before being used for lipid separation. Iodine vapor was applied to TLC plates after lipid separation for brief reversible staining. Different lipid groups on the TLC plates were marked with a pencil and were scraped for analysis. For LC/MS analysis, lipids were extracted using the buffer containing acetonitrile/isopropanol/water (3:3:2) with 0.1% formic acid and 10 μM propyl-4-hydroxybenzoate as the internal standard.

## Protein expression and ACS enzyme assay

To express His-tagged recombinant protein Sl-AACS1, the full-length *Sl-AACS* ORF sequence was amplified using the primer pair SlAACS1-pET28_F/R (*Supplementary file 3*) and was cloned into pET28b (EMD Millipore) using the Gibson assembly kit (NEB). The pET28b vector was linearized by digesting with BamHI and XhoI to create overhangs compatible for Gibson assembly. The pET28b constructs were transformed into BL21 Rosetta cells (EMD Millipore). The protein expression was induced by adding 0.05 mM isopropyl $\beta$-D-1-thiogalactopyranoside to the cultures when the $OD_{600nm}$ = 0.5. The *E. coli* cultures were further grown overnight at 16 °C, 120 rpm. The His-tagged

proteins were purified by Ni-affinity gravity-flow chromatography using the Ni-NTA agarose (Qiagen) following the product manual.

Measurement of acyl-CoA synthetase activity was performed using minor modifications of the coupled enzyme assay described by *Schneider et al., 2005*. A multimode plate reader (PerkinElmer, mode EnVision 2104) compatible with the 96-well UV microplate was used for the assays. The fatty acid substrates were dissolved in 5% Triton X-100 (v/v) to make 5 mM stock solutions. The enzyme assay premix was prepared containing 0.1 M Tris-HCl (pH 7.5), 2 mM dithiothreitol, 5 mM ATP, 10 mM $MgCl_2$, 0.5 mM CoA, 0.8 mM NADH, 250 µM fatty acid substrate, 1 mM phosphoenolpyruvate, 20 units myokinase (Sigma-Aldrich, catalog no. M3003), 10 units pyruvate kinase (Roche, 10128155001), 10 units lactate dehydrogenase (Roche, 10127230001), and was aliquoted 95 µL each to the 96-well microplate. The reaction was started by adding 5 µL (1–2 µg) proteins. The chamber of the plate reader was set to 30 ˚C and the OD at 340 nm was recorded every 5 min for 40 min. Oxidation of NADH, which is monitored by the decrease of $OD_{340nm}$, was calculated using the NADH extinction coefficient 6.22 $cm^2$ $µmol^{-1}$. Every two moles of oxidized NADH is equivalent to one mole of acyl-CoA product generated in the reaction. To measure the parameters of enzyme kinetics, the fatty acid substrate concentration was varied from 0 to 500 µM, with NADH set at 1 mM. The fatty acid substrates, sodium acetate (C2:0), sodium butyrate (C4:0), sodium hexanoate (C6:0), sodium octanoate (C8:0), sodium decanoate (C10:0), sodium laurate (C12:0), sodium myristate (C14:0), sodium palmitate (C16:0), and sodium stearate (C18:0), were purchased from Sigma-Aldrich. *Trans*-2-decenoic acid (C10:1), 8-methylnonanoic acid (iC10:0), 3-hydroxy lauric acid (C12:OH), and 3-hydroxy myristic acid (C14:OH) were purchased from Cayman Chemical.

## RNA extraction, sequencing, and differential gene expression analysis

Total RNA was extracted from trichomes isolated from stems and shaved stems of 7-week-old *S. pennellii* LA0716 plants using the RNAeasy Plant Mini kit (Qiagen) and digested with DNase I. A total of four RNA samples extracted from two tissues with two replicates were used for RNA sequencing. The sequencing libraries were prepared using the KAPA Stranded RNA-Seq Library Preparation Kit. Libraries went through quality control and quantification using a combination of Qubit dsDNA high sensitivity (HS), Applied Analytical Fragment Analyzer HS DNA and Kapa Illumina Library Quantification qPCR assays. The libraries were pooled and loaded onto one lane of an Illumina HiSeq 4000 flow cell. Sequencing was done in a 2 × 150 bp paired end format using HiSeq 4000 SBS reagents. Base calling was done by Illumina Real Time Analysis (RTA) v2.7.6 and output of RTA was demultiplexed and converted to FastQ format with Illumina Bcl2fastq v2.19.1.

The paired end reads were filtered and trimmed using Trimmomatic v0.32 (*Bolger et al., 2014a*) with the setting (LLUMINACLIP: TruSeq3-PE.fa:2:30:10 LEADING:3 TRAILING:3 SLIDING-WINDOW:4:30), and then mapped to the *S. pennellii* LA0716 genome v2.0 (*Bolger et al., 2014a*) using TopHat v1.4 (*Trapnell et al., 2009*) with the following parameters: -p (threads) 8, -i (minimum intron length) 50, -I (maximum intron length) 5000, and -g (maximum hits) 20. The FPKM (Fragments Per Kilobase of transcript per Million mapped reads) values for the genes were analyzed via Cufflinks v2.2 (*Trapnell et al., 2010*).For differential expression analysis, the HTseq package (*Anders et al., 2015*) in Python was used to get raw read counts, then Edge R version 3.22.5 (*McCarthy et al., 2012*) was used to compare read counts between trichome-only RNA and shaved stem RNA using a generalized linear model (glmQLFit).

## VIGS and qRT-PCR

For VIGS analysis of *Sq-AACS1* and *Sq-AECH1* in *S. quitoense*, the fragments of these two genes, as well as the phytoene desaturase (PDS) gene fragment, were amplified using the primers listed in *Supplementary file 3*, cloned into pTRV2-LIC (ABRC no. CD3-1044) using the ligation-independent cloning method (*Dong et al., 2007*), and transformed into *A. tumefaciens* (strain GV3101). The VIGS experiments were performed as described (*Leong et al., 2020*). In brief, the Agrobacterium strains harboring pTRV2 constructs, the empty pTRV2, or pTRV1 were grown overnight in separate LB cultures containing 50 µg/mL kanamycin, 10 µg/mL gentamicin, and 50 µg/ml rifampicin at 30 ˚C. The cultures were re-inoculated in the induction media (50 mM MES pH5.6, 0.5% glucose [w/v], 2 mM $NaH_2PO_4$, 200 µM acetosyringone) for overnight growth. The cells were harvested, washed, and resuspended in the buffer containing 10 mM MES, pH 5.6, 10 mM $MgCl_2$, and 200 µM

acetosyringone with the $OD_{600nm}$ = 1. Different cultures containing pTRV2 constructs were mixed with equal volume of pTRV1 cultures prior to infiltration. The 2- to 3-week-old young *S. quitoense* seedlings grown under 16 hr photoperiod at 24°C were used for infiltration: the two fully expanded cotyledons were infiltrated. Approximately three weeks post inoculation, the fourth true leaf of each infiltrated plant was cut in half for acylsugar quantification and gene expression analysis, respectively. The onset of the albino phenotype of the control group infiltrated with the PDS construct was used as a visual marker to determine the harvest time and leaf selection for the experimental groups. At least fourteen plants were analyzed for each construct. The trichome acylsugars were extracted using the solution containing acetonitrile/isopropanol/water (3:3:2) with 0.1% formic acid and 1 μM telmisartan as internal standard, following the protocol at http://dx.doi.org/10.17504/protocols.io. xj2fkqe.

The leaf RNA was extracted using RNeasy Plant Mini kits (Qiagen) and digested with DNase I. The first-strand cDNA was synthesized by Superscript II (Thermofisher Scientific) using total RNA as templates. Quantitative real-time PCR was performed to analyze the *Sq-AACS1* or *Sq-AECH1* mRNA in *S. quitoense* leaves using the primers listed in *Supplementary file 3*. EF1α was used as a control gene. A QuantStudio 7 Flex Real-Time PCR System with Fast SYBR Green Master Mix (Applied Biosystems) was used for the analysis. The relative quantification method ($2^{-\Delta\Delta Ct}$) was used to evaluate the relative transcripts levels.

## LC/MS analysis

Trichome acylsugars extracted from tomato IL and BILs were analyzed using a Shimadzu LC-20AD HPLC system connected to a Waters LCT Premier ToF mass spectrometer. Ten microliter samples were injected into a fused core Ascentis Express C18 column (2.1 mm ×10 cm, 2.7 μm particle size; Sigma-Aldrich) for reverse-phase separation with column temperature set at 40°C. The starting condition was 90% solvent A (0.15% formic acid in water) and 10% solvent B (acetonitrile) with flow rate set to 0.4 mL/min. A 7 min linear elution gradient was used: ramp to 40% B at 1 min, then to 100% B at 5 min, hold at 100% B to 6 min, return to 90% A at 6.01 min and hold until 7 min.

For analyzing trichome acylsugars extracted from *S. pennellii* transgenic plants and membrane lipids from *N. benthamiana*, a Shimadzu LC-20AD HPLC system connected to a Waters Xevo G2-XS QToF mass spectrometer was used. The starting conditions were 95% solvent A (10 mM ammonium formate, pH 2.8) and 5% solvent B (acetonitrile) with flow rate set to 0.3 mL/min. A 7 min linear elution gradient used for acylsugar analysis was: ramp to 40% B at 1 min, then to 100% B at 5 min, hold at 100% B to 6 min, return to 95% A at 6.01 min and hold until 7 min. A 12 min linear elution gradient used for the lipid analysis was: ramp to 40% B at 1 min, then to 100% B at 5 min, hold at 100% B to 11 min, return to 95% A at 11.01 min and hold until 12 min.

For analyzing trichome acylsugars extracted from other plants, a Waters Acquity UPLC was coupled to a Waters Xevo G2-XS QToF mass spectrometer. The starting condition was 95% solvent A (10 mM ammonium formate, pH 2.8) and 5% solvent B (acetonitrile) with flow rate set to 0.3 mL/min. A 7 min linear elution gradient was: ramp to 40% B at 1 min, then to 100% B at 5 min, hold at 100% B to 6 min, return to 95% A at 6.01 min and held until 7 min. A 14 min linear elution gradient was: ramp to 35% B at 1 min, then to 85% B at 12 min, then to 100% B at 12.01 min, hold at 100% B to 13 min, return to 95% A at 13.01 min and held until 14 min.

For Waters LCT Premier ToF mass spectrometer, the MS settings of electrospray ionization in negative mode were: 2.5 kV capillary voltage, 100°C source temperature, 350°C desolvation temperature, 350 liters/h desolvation nitrogen gas flow rate, 10 V cone voltage, and mass range *m/z* 50 to 1500 with spectra accumulated at 0.1 s/function. Three collision energies (10, 40, and 80 eV) were used in separate acquisition functions to generate both molecular ion adducts and fragments. For Waters Xevo G2-XS QToF mass spectrometer, the MS settings of the negative ion-mode electrospray ionization were as follows: 2.00 kV capillary voltage, 100°C source temperature, 350°C desolvation temperature, 600 liters/h desolvation nitrogen gas flow rate, 35 V cone voltage, mass range of *m/z* 50 to 1000 with spectra accumulated at 0.1 s/function. Three collision energies (0, 15, and 35 eV) were used in separate acquisition functions. The MS settings for positive ion-mode electrospray ionization were: 3.00 kV capillary voltage, 100°C source temperature, 350°C desolvation temperature, 600 liters/h desolvation nitrogen gas flow rate, 35 V cone voltage, mass range of *m/z* 50 to 1000 with spectra accumulated at 0.1 s/function. Three collision energies (0, 15, and 45 eV) were used in separate acquisition functions. The Waters QuanLynx software was used to integrate peak

areas of the selected ion relative to the internal standard. For quantification purpose, data collected with the lowest collision energy was used in the analysis.

## Gene coexpression analysis

The publicly available tomato RNA-seq datasets and the methods used for normalizing FPKM, gene expression correlation analysis were described in a recent study (*Moore et al., 2020*). 926 RNA-seq Sequence Read Archive (SRA) files for tomato from 47 studies were downloaded from National Center for Biotechnology Information (NCBI; https://www.ncbi.nlm.nih.gov/) (Table S6 in *Moore et al., 2020*). Reads were filtered using Trimmomatic (*Bolger et al., 2014b*) based on the sequence quality with default settings, and mapped to the tomato NCBI *S. lycopersicum* genome 2.5 using TopHat (*Trapnell et al., 2009*). Read files with <70% mapped reads were discarded. Fragments per kilobase of transcript per million mapped reads (FPKM) were calculated using Cufflinks (*Trapnell et al., 2010*). The pipeline for FPKM calling used in this study was put in https://github.com/ShiuLab/RNA-seq_pipeline (*Uygun et al., 2020*; copy archived at https://github.com/elifesciences-publications/RNAseq_pipeline). The median FPKM of multiple replicates was used for each sample, resulting in FPKM values in 372 samples. To draw the heatmap of gene expression profiles, FPKM values of a gene across all the samples were scaled, where the maximum FPKM was scaled to 1, while the minimum value was 0.

## Synteny scan

Protein sequences of annotated genes and the corresponding annotation files in General Feature Format (GFF) of 11 Solanaceae species, *Ipomoea trifida*, and *Coffea canephora* were downloaded from National Center for Biotechnology Information (NCBI, https://www.ncbi.nlm.nih.gov/genome/) or Solanaceae Genomics Network (SGN, https://solgenomics.net/). The GFF files contain the coordinates of annotated genes on assembled chromosomes or scaffolds. The sources and version numbers of sequences and GFF files used are: *S. lycopersicum* ITAG3.2 (SGN) and V2.5 (NCBI), *S. pennellii* SPENNV200 (NCBI) and v2.0 (SGN), *S. tuberosum* V3.4 (SGN), *S. melongena* r2.5.1 (SGN), *Capsicum annuum L. zunla-1* V2.0 (SGN), *C. annuum_var. glabriusculum* V2.0 (SGN), *Nicotiana attenuata* NIATTr2 (SGN), *N. tomentosiformis* V01 (NCBI), *N. benthamiana* V1.0.1 (SGN), *Petunia axillaris* V1.6.2 (SGN), *P. inflata* V1.0.1 (SGN), *I. trifida* V1.0 (NCBI), and *C. canephora* Vx (SGN).

To hypothesize the evolutionary history of genes in the acylsugar gene cluster, putative pseudogenes, which are homologs to protein-coding genes but with predicted premature stops/frameshifts and/or protein sequence truncation, were also identified for each species as described (*Wang et al., 2018*). Protein sequences from *Arabidopsis thaliana*, *Oryza sativa*, and *S. lycopersicum* were used as queries in the searches against the genomic regions of target species using TBLASTN (*Altschul et al., 1990*). The intergenic genomic sequences were identified as potential pseudogenes using the pipeline from as previously described (*Campbell et al., 2014*; *Zou et al., 2009*). If one of the six-frame translated sequences of the intergenic genomic sequences had significant similarity to annotated protein sequences, and had premature stops/frameshifts and/or were truncated (<30% of functional paralogs), the gene was defined as a pseudogene.

Genome-wide syntenic analysis was conducted using annotated protein-coding genes and putative pseudogenes from all the species with MCScanX-transposed (*Wang et al., 2013*) as described (*Wang et al., 2018*). The MCScanX-based analysis did not lead to a syntenic block of acylsugar gene cluster on chromosome 7 of *S. melongena* r2.5.1, which can be due to true absence, issues with genome assembly, or lack of coverage. To verify this, protein sequences of *S. lycopersicum* genes in genomic blocks on chromosome 7 were searched against an updated *S. melongena* genome from The Eggplant Genome Project (http://ddlab.dbt.univr.it/eggplant/) that led to the identification of the synteny.

## Phylogenetic tree building

Homologous genes of *Sl-AACS1* (ACS), *Sl-AECH1* (ECH), and *Solyc07g043670* (BAHD acyltransferase) were obtained through BLAST (*Altschul et al., 1990*) search from the genomes of 11 Solanaceae species, *Ipomoea trifida*, and *Coffea canephora* with an Expect value threshold of 1e-5. To simply the phylogenetic tree, sequences which are distantly related to the target genes were removed, and the remained sequences were used to rebuild the phylogenetic trees. The amino acid

sequences were aligned using MUSCLE (*Edgar, 2004*) with the default parameters. The phylogenetic trees were built using the maximum likelihood method with 1000 bootstrap replicates. The trees were generated using RAxML/8.0.6 (*Stamatakis, 2014*) with the following parameters: -f a -x 12345 p 12345 -# 1000 m PROTGAMMAAUTO –auto-prot=bic, and were shown with midpoint rooting. The final sequence alignments used to generate the phylogenetic trees were provided in *Supplementary file 5*.

### Ancestral trait reconstruction

Ancestral trait state reconstruction was conducted using the maximum likelihood model Mk1 in Mesquite 3.6 (*Massidon and Maddison, 2018*). Four traits were inferred for their ancestral states. They are the presence of medium chain acylsugars, presence of ACS genes in the synteny, presence of ECH genes in the synteny, and presence of both ACS and ECH genes in the synteny. The phylogeny of Solanaceae species was based on a previous study (*Särkinen et al., 2013*).

### Acylsugar acyl chain composition analysis by GC-MS

Acyl chains were characterized from the corresponding fatty acid ethyl esters following transesterification of acylsugar extractions as previously reported (*Ning et al., 2015*). Plants were grown for 4–8 weeks and approximately ten leaves were extracted for 3 min in 10 mL of 1:1 isopropanol:acetonitrile with 0.01% formic acid. Extractions were dried to completeness using a rotary evaporator and then 300 µL of 21% (v/v) sodium ethoxide in ethanol (Sigma) was added and incubated for 30 min with gentle rocking and vortexing every five minutes and 400 µL hexane was added and vortexed for 30 s. To the hexane layer, 500 µL of saturated sodium chloride in water was added and vortexed to facilitate a phase separation. After phase separation, the top hexane layer was transferred to a new tube. The phase separation by addition of 500 µL hexane was repeated twice, with the final hexane layer transferred to a 2 mL glass vial with a glass insert.

The fatty acid ethyl esters were analyzed using an Agilent 5975 single quadrupole GC-MS equipped with a 30 m, 0.25 mm internal diameter fused silica column with a 0.25 µm film thickness VF5 stationary phase (Agilent). Injection of 1 µL of each hexane extract was performed using splitless mode. The gas chromatography program was as follows: inlet temperature, 250˚C; initial column temperature, 70˚C held for 2 min; ramped at 20 ˚C/min until 270˚C, then held at 270˚C 3 min. The helium carrier gas was used with 70 eV electron ionization. Acyl chain type was determined through NIST Version 2.3 library matches of the mass spectra of the corresponding ethyl ester and relative abundances were determined through integrating the corresponding peak area over the total acyl chain peak area.

## Acknowledgements

We thank the CM Rick Tomato Genetics Resource Center (University of California Davis, CA USA) for providing tomato seeds, Zamir lab in Hebrew University of Jerusalem for providing tomato ILs and BILs seeds. We acknowledge Dr. Kun Wang and Dr. Christoph Benning for their helpful guidance in lipid analysis. We thank Krystle Wiegert-Rininger and Cornelius Barry for their help in RNA sequencing and Dr. Kent Chapman from University of North Texas for helpful discussions. We acknowledge Kathleen Imre and Sara Haller for their help with tomato transformation. We thank the MSU Center for Advanced Microscopy and RTSF Mass Spectrometry and Metabolomics Core Facilities for their support with LC/MS analysis.

## Additional information

### Funding

| Funder | Grant reference number | Author |
|---|---|---|
| National Science Foundation | 1546617 | Shin-Han Shiu Robert L Last |
| National Science Foundation | 1655386 | Shin-Han Shiu |
| U.S. Department of Energy | BER DE-SC0018409 | Shin-Han Shiu |

| National Science Foundation | 1727362 | Federica Brandizzi |
| --- | --- | --- |
| National Institutes of Health | GM110523 | Bryan J Leong<br>Robert L Last |
| National Science Foundation | 1757043 | Rachel Combs<br>Robert L Last |
| National Science Foundation | 1811055 | Craig A Schenck |

The funders had no role in study design, data collection and interpretation, or the decision to submit the work for publication.

### Author contributions
Pengxiang Fan, Conceptualization, Data curation, Formal analysis, Supervision, Investigation, Visualization, Methodology, Writing - original draft, Writing - review and editing; Peipei Wang, Formal analysis, Visualization, Methodology, Writing - original draft, Writing - review and editing; Yann-Ru Lou, Conceptualization, Formal analysis, Writing - review and editing; Bryan J Leong, Conceptualization, Methodology, Writing - review and editing; Bethany M Moore, Craig A Schenck, Formal analysis, Writing - review and editing; Rachel Combs, Formal analysis, Methodology, Writing - review and editing; Pengfei Cao, Federica Brandizzi, Formal analysis, Visualization, Methodology, Writing - review and editing; Shin-Han Shiu, Formal analysis, Funding acquisition, Writing - original draft, Writing - review and editing; Robert L Last, Conceptualization, Supervision, Funding acquisition, Methodology, Writing - original draft, Project administration, Writing - review and editing

### Author ORCIDs
Pengxiang Fan (iD) https://orcid.org/0000-0002-4560-3783
Peipei Wang (iD) https://orcid.org/0000-0002-7580-9627
Yann-Ru Lou (iD) https://orcid.org/0000-0002-4716-4323
Bryan J Leong (iD) http://orcid.org/0000-0003-4042-1160
Bethany M Moore (iD) https://orcid.org/0000-0002-2104-7292
Craig A Schenck (iD) http://orcid.org/0000-0002-5711-7213
Rachel Combs (iD) http://orcid.org/0000-0001-6626-0903
Pengfei Cao (iD) http://orcid.org/0000-0001-6998-9302
Federica Brandizzi (iD) http://orcid.org/0000-0003-0580-8888
Shin-Han Shiu (iD) https://orcid.org/0000-0001-6470-235X
Robert L Last (iD) https://orcid.org/0000-0001-6974-9587

### Decision letter and Author response
Decision letter https://doi.org/10.7554/eLife.56717.sa1
Author response https://doi.org/10.7554/eLife.56717.sa2

## Additional files

### Supplementary files
• Supplementary file 1. Co-expression analysis of tomato genes from ACS, ECH, and BAHD acyltransferase families used for phylogenetic analysis in this study. The values of Pearson's correlation coefficient of the expression profiles between any of the two genes were shown in the table. The coefficient values were generated using the FPKM values of these genes in the 372 RNA-seq samples as shown in *Figure 2—figure supplement 1*. The orange box highlights a group of co-expressed genes involved in acylsugar biosynthesis, such as *Sl-ASATs*, *Sl-AACS1*, and *Sl-AECH1*. The purple box points out another group of co-expressed genes that are root hair specific.

• Supplementary file 2. Gene expression levels of all analyzed transcripts in *Solanum pennellii* LA0716. logFC: log2 fold change in stem trichomes versus shaved stems. logCPM: log (counts per million) in trichomes versus shaved stems. The F and Q-value test the significance of differential expression via a quasi- general linear model. The values noted in the sample columns represent the FPKM (Fragments Per Kilobase of transcript per Million mapped reads) analyzed via Cufflinks.

- Supplementary file 3. Synthesized gene fragments and primers used in this study.
- Supplementary file 4. The date used to generate the synteny figure shown in *Figure 6—figure supplement 1*.
- Supplementary file 5. The sequence alignment documents used to generate the phylogenetic trees for *Figure 6—figure supplements 2*, *3*, *4*, *5* and *6*.
- Transparent reporting form

## Data availability

The RNA-seq reads were deposited in the National Center for Biotechnology Information Sequence Read Archive under the accession number PRJNA605501. Sequence data used in this study are in the GenBank/EMBL data libraries under these accession numbers: Sl-AACS1(MT078737), Sl-AECH1 (MT078736), Sp-AACS1(MT078735), Sp-AECH1(MT078734), Sq-AACS1(MT078732), Sq-AECH1 (MT078731), Sq_c35719 (MT078733). The following materials require a material transfer agreement: pEAQ-HT, pK7WG, pKGWFS7, pEarleyGate102, pEarleyGate104, pTRV2-LIC, pICH47742::2x35S-5'UTR-hCas9(STOP)-NOST, pICH41780, pAGM4723, and pICSL11024.

The following dataset was generated:

| Author(s) | Year | Dataset title | Dataset URL | Database and Identifier |
|---|---|---|---|---|
| Fan P, Last RL | 2020 | Solanum pennellii stem and trichome transcriptome | https://www.ncbi.nlm.nih.gov/bioproject/PRJNA605501 | NCBI BioProject, PRJNA605501 |

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
