## [Decision Letter]

**Acceptance summary:**

Modern genomics has greatly accelerated the ability to identify specialized metabolite pathways moving the research from simple cataloging to begin asking deeper questions about how novelty evolves. In this work, the authors begin to study the evolution of a gene cluster within a plant family. This illustrates the complexity that will begin to be uncovered when these studies are more widely conducted.

**Decision letter after peer review:**

Thank you for submitting your article "Evolution of a plant gene cluster in Solanaceae and emergence of metabolic diversity" for consideration by *eLife*. Your article has been reviewed by three peer reviewers, and the evaluation has been overseen by a Reviewing Editor and Christian Hardtke as the Senior Editor. The following individuals involved in review of your submission have agreed to reveal their identity: Reuben Peters (Reviewer #3).

The reviewers have discussed the reviews with one another and the Reviewing Editor has drafted this decision to help you prepare a revised submission.

As the editors have judged that your manuscript is of interest, but as described below that additional experiments are required before it is published, we would like to draw your attention to changes in our revision policy that we have made in response to COVID-19 (https://elifesciences.org/articles/57162). First, because many researchers have temporarily lost access to the labs, we will give authors as much time as they need to submit revised manuscripts. We are also offering, if you choose, to post the manuscript to bioRxiv (if it is not already there) along with this decision letter and a formal designation that the manuscript is 'in revision at *eLife*'. Please let us know if you would like to pursue this option. (If your work is more suitable for medRxiv, you will need to post the preprint yourself, as the mechanisms for us to do so are still in development.)

Summary and essential revisions:

This work looks into the potential evolution of a set of genes within the Solanaceae using a blend of biochemistry and phylogenetics.

In the revision, please:

1) Address the WGD vs other forms of structural variation (tandem duplication, duplication and distal movement, etc.) topics raised by the reviewers.

2) Describe the gene selection and include all possible genes that may be evolutionarily linked as indicated by reviewer 1.

3) Enhance the Introduction and Discussion to provide a broader view of clustered vs non-clustered pathways in plant specialized metabolism.

Reviewer #1:

The authors provide strong evidence that the two genes are specifically expressed in the trichome tip cells and are involved in production of medium chain acylsugars. However, I found the comparative genomic analyses to be confusing. More work is necessary to support the authors' evolutionary model.

1) Evolutionary analysis

1a) The Figure 6 model is too simplistic to follow the evolutionary history of all the gene duplicates in the region, while the Figure 6 supplemental synteny plot is too small to interpret. In the supplemental, I can't tell which genes are ECHs, ACSs, etc. because the rectangles are too small. There are many ACS, BAHD, and ECH paralogs in this syntenic block and the evolutionary history of all these paralogs is unclear. Moreover, by focusing exclusively on the co-located genes, the authors miss an opportunity to evaluate the evolutionary history of these two (three?) genes more globally. Where/how did the ancestral AACS1 and AECH1 genes originate? Was there genomic rearrangment? Non tandem gene duplication? An expanded phylogenetic analysis would be very helpful here. An OrthoFinder analysis on the 13 genomes included in Figure 6, would quickly provide the raw gene families for these phylogenies. This would allow the authors to elaborate on when the hijacking of primary metabolism occurred (Discussion paragraph one). Are AACS1 and AECH1 grouping sister to genes involved in primary metabolism? Or are they grouping with additional homologs of unknown or specialized function?

1b) There are no methods provided for how the sequences were selected for the Figure 4—figure supplement 2C phylogeny and I didn't see any methods for alignment or tree construction either. The ECH-containing orthogroup on PLAZA (https://bioinformatics.psb.ugent.be/plaza/versions/plaza_v4_dicots/) contain several other ECH homologs (Solyc07g043690.1, Solyc07g044730.2, Solyc07g044710.1, Solyc07g044720.1) that appear to be grouping between AECH1 and Solyc06g54520, but these sequences are not included in the supplemental phylogeny.

2) Framing

2a) The authors mention in the Abstract and again in the Introduction that specialized metabolic gene clusters (SMGCs) are a hallmark of plant specialized metabolism. This gives the impression that gene clusters in plants are more common than they are. In fungal and bacterial genomes, gene clusters predominate. In plants, the pattern is not nearly as straight forward. Many plant specialized pathways are not clustered, and those pathways that are clustered are only partially so (i.e. these clusters are missing pathway regulators, product transporters, additional accessory enzymes, etc., which are common in microbial clusters); see Wisecaver et al., 2017. Given the extensive literature on complex microbial SMGCs, referring to two co-located genes as a SMGC feels like a bit of a stretch without additional clarification. Sometimes the authors appear to include the BAHD in the gene cluster, sometimes it's referred to as a two gene cluster. I wasn't sure whether BAHD was perhaps already know to function earlier in the pathway? The gene wasn't differentially expressed in the trichomes, so is it not involved? If no, then why is it included in the model?

2b) Similarly, I question the use of the term supercluster here as well. The fumagillin and pseurotin superclusters in *Aspergillus* are at least 29 genes long, co-regulated, intertwined, and maintained in diverse *Aspergillus* species despite being in the telomeric region of otherwise rapidly rearranging genomes (Wiemann et al., 2013). Do the authors believe that the co-location of these genes is being maintained in *Solanum* and are the genes being co-regulated?

Reviewer #2:

General assessment. The research is similar in scope to a previous *eLife* publication by the Last lab (Moghe et al., 2017). I assume the biochemical and plant transformation experiments are valid; however, this is not my area of expertise, so I focus my comments on the comparative genomic and evolutionary claims.

1) Novelty and framework for discussing innovation in metabolism with respect to clusters and “superclusters.” The novelty of the manuscript (for researchers who work outside Solanaceae and trichome biology) lies in the final paragraph discussing how this may be the first paper to describe "superclusters." The Abstract, Introduction and Discussion could all benefit by citing more literature that encompasses the ongoing debates about the prevalence of "clusters" or not. The authors ask this as a question in the final paragraph; however this has been addressed in research not cited here (for example, Wisecaver et al., 2017). As the paper currently reads, they leave off most research (aside from their own self citations) that is outside of the "cluster paradigm" of the Osbourn lab (e.g., Nutzmann et al., 2016). It would be interesting to know if a "Wisecaver-type network analysis" would get similar results.

2) Gene and genome duplication analyses. The authors discuss evolution and gene duplication; however, they do not use the genomic resources of the 13 species examined to phylogenetically evaluate whether the duplicates that may be involved in the phenotype are from whole genome duplication (WGD or polyploidy) vs. small scale duplication (SSD). Given that Solanaceae have a WGD (as a triplication shown in Figure 6A), it would seem important to investigate the acylsugar biosynthesis pathway in this context (and any inferred patterns of loss after triplication if now in single or duplicate copy).

3) Experimental Design. A species phylogeny is given in Figure 6B for the species analyzed and the corresponding results and discussion hypothesize gain and loss events. Do the authors use a formal ancestral reconstruction analysis to back up their evolutionary scenario or is it just using parsimony to explain biochemical observations? There are formal tools used by evolutionary biologists to infer trait evolution that could be used here. This would seem particularly relevant for any claims made about the other genera (*Jaltomata*, *Physalis*, *Iochroma*, *Atropa*, and *Hyoscyamus*) that have missing data. It is not required; however, there other resources for other genera and sister families that could be exploited in a transcriptome analysis (e.g. One Thousand Plant Transcriptomes Initiative. 2019. One thousand plant transcriptomes and phylogenomics of green plants. Nature 574: 679-685).

Otherwise, this is an excellent manuscript.

Reviewer #3:

This manuscript describes investigation of biosynthesis of medium chain length acylsugars in the Solanaceae, involving identification of an intriguing biosynthetic gene cluster. In particular, two new enzymes are identified and characterized here, which were found to be part of a complex genetic loci containing three different types of enzymes, each of which has undergone multiple tandem gene duplication. However, those for the “other” (third) enzyme in the original (chromosome 7) cluster are not required for this medium chain acylsugar biosynthesis, but rather a paralog on chromosome 12 instead. Intriguingly, this cluster is adjacent to the larger portion of the split cluster previously identified for steroidal alkaloid biosynthesis. It is tempting to speculate that assembly, with subsequent splitting, occurred together. Although the authors have avoided this more speculative hypothesis, it might be of interest to look at conservation of the split steroidal alkaloid cluster, analogous to that already reported here for the acylsugar biosynthetic cluster, to examine this hypothesis. In addition, one of the biochemical findings reported here is somewhat puzzling. Specifically, the characterized activity of the acylsugar acyl-CoA synthetase (AACS) seems to be higher with short, particularly C8, rather than medium (>C10) chain length fatty acids. The authors should at least note this in the Discussion. Otherwise the work is solid, the results interesting and well-presented.

[Editors' note: further revisions were suggested prior to acceptance, as described below.]

Thank you for resubmitting your work entitled "Evolution of a plant gene cluster in Solanaceae and emergence of metabolic diversity" for further consideration by *eLife*. Your revised article has been evaluated by Christian Hardtke (Senior Editor) and a Reviewing Editor.

The manuscript has been improved but there are some remaining issues that need to be addressed before acceptance, as outlined below:

Please see the few questions by reviewer 1 that need editorial clarification within the manuscript. Other readers will likely have the same thoughts and any effort to address these questions would be greatly helpful to the future use of the paper by the community.

Reviewer #1:

This revision satisfies most of my comments on the previous draft. The evolutionary analysis of the three gene families is much improved.

Could the segmental duplication of acyl-CoA have occurred in an ancestor of *Solanum* and *Nicotiana* following the divergence from *Petunia*? That seems like the most parsimonious explanation. What am I missing?

In the response to my earlier comment, the authors say that understanding of the mechanism of hijacking primary metabolism for acylsugar biosynthesis requires future work. However, the manuscript still reports to provide insights into this co-option in several places. Can the authors tone down these statements or make it more explicit what they mean here?

I think the point of the co-expression analysis was missed a bit by focusing only on other ACS, ECH, and BAHDs. It looks like the Moore et al., 2020 study called modules of co-expressed genes using several different metrics, while here the authors only report the pairwise coexpression between ACS, ECH, and BAHDs without binning these genes into discrete modules or looking for associations with other gene families. This may be beyond the scope of this manuscript. However, I think it is definitely something worth investigating at some point to better understand the evolution of the larger metabolic pathway rather than the portion contained within this genomic region.

Reviewer #2:

This is a resubmission of a manuscript that I previously reviewed (Reviewer #2). The authors conducted the additional analyses that the external reviewers requested (e.g., co-expression analyses, updated ancestral state reconstruction) and added references and addressed issues raised. I appreciate that the authors found errors in their previous analyses and dropped the *Nicotiana attenuata* AcS gene result given what the updated syntenic analyses found. I was slightly disappointed that the authors have still not been able to clearly disentangle the impact of whole genome duplications versus tandem duplications on this pathway (and the order of what may be nested tandem duplications) or did not find any genomic footprints for putative losses; however, this is likely due to incomplete genome sequence and taxon sampling and thus beyond the scope of what can be inferred at this time. As such, I have no further significant recommendations to improve the manuscript with respect to analyses.

The only substantial issue is that is it not clear to me what a "super-cluster" is or why that new terminology is needed here. The response to reviewers was clearer than the revised manuscript about this issue (relative to what is seen in fungi as noted by another reviewer).

In summary, this is another rare but valuable example of a duplicated gene cluster leading to novelty in specialized metabolism.

Reviewer #3:

This revised manuscript addresses all of my concerns.

---

## [Author Response]

Summary and essential revisions:This work looks into the potential evolution of a set of genes within the Solanaceae using a blend of biochemistry and phylogenetics.In the revision, please:1) Address the WGD vs other forms of structural variation (tandem duplication, duplication and distal movement, etc.) topics raised by the reviewers.2) Describe the gene selection and include all possible genes that may be evolutionarily linked as indicated by reviewer 1.3) Enhance the Introduction and Discussion to provide a broader view of clustered vs non-clustered pathways in plant specialized metabolism.

As detailed in our response to the reviewer comments, we have made the following changes to the manuscript.

- The gene phylogenetic analysis was performed to better understand the evolutionary history of ACS, ECH, and BAHD acyltransferase in the syntenic regions.

- Gene co-expression analysis was done using ACS, ECH, and BAHD acyltransferase family genes to test whether Sl-AACS1 and Sl-AECH co-express with other acylsugar related genes.

- Trait ancestral state reconstruction analysis was carried out using the likelihood method.

- We revised the abstract, introduction, and discussion to provide a broader view of clustered specialized metabolism genes.

- During the phylogenetic analysis, we found that the *Nicotiana attenuata* ACS gene NIATv7_g15235 is distantly related to other ACS genes. We revisited the results of synteny analysis and found that the *N. attenuata* region containing NIATv7_g15235 was incorrectly identified as a synteny of the Chr7/Chr12 region in our previous draft. This *Nicotiana* region is distinct from the synteny on Chr7/Chr12 and was removed from the revised paper. The previous supplemental file1 related to the analysis of NIATv7_g15235 was also removed.

Reviewer #1:[…]1) Evolutionary analysis1a) The Figure 6 model is too simplistic to follow the evolutionary history of all the gene duplicates in the region, while the Figure 6 supplemental synteny plot is too small to interpret. In the supplemental, I can't tell which genes are ECHs, ACSs, etc. because the rectangles are too small. There are many ACS, BAHD, and ECH paralogs in this syntenic block and the evolutionary history of all these paralogs is unclear. Moreover, by focusing exclusively on the co-located genes, the authors miss an opportunity to evaluate the evolutionary history of these two (three?) genes more globally. Where/how did the ancestral AACS1 and AECH1 genes originate? Was there genomic rearrangment? Non tandem gene duplication? An expanded phylogenetic analysis would be very helpful here. An OrthoFinder analysis on the 13 genomes included in Figure 6, would quickly provide the raw gene families for these phylogenies. This would allow the authors to elaborate on when the hijacking of primary metabolism occurred (Discussion paragraph one). Are AACS1 and AECH1 grouping sister to genes involved in primary metabolism? Or are they grouping with additional homologs of unknown or specialized function?

1) The Figure 6-supplemental figure with the synteny plot was improved. Large arrows with three different colors were used in the figure to point out the genes from the three families. The raw data with gene ID and location information used to generate this figure was included as a supplementary file.

2) We performed phylogenetic analysis to better understand the evolutionary history of ACS, ECH, and BAHD acyltransferase in the syntenic regions and provided a clearer picture of how these orthologs/paralogs involved in acylsugar biosynthesis evolved. Homologous genes of Sl-AACS1 (ACS), Sl-AECH1(ECH), and Solyc07g043670 (BAHD acyltransferase) were obtained through BLAST from the genomes of 13 species and were used to build three phylogenetic trees as shown in Figure 6—figure supplement 2, 3, and 4. The evolutionary history and duplication mechanisms that gave rise to the genes in the syntenic regions of different Solanaceae species were inferred. The phylogenetic analysis supports our proposed evolution model regarding the temporal order of emergence of the three types of genes in the syntenic region.

3) After this more detailed analysis, both AACS1 and AECH1 group sister to additional homologs with uncharacterized functions. Thus, future efforts will be required to further our understanding of the mechanism of “hijacking” primary metabolism for acylsugar biosynthesis. The Discussion was modified to reflect the new phylogenetic analysis results.

1b) There are no methods provided for how the sequences were selected for the Figure 4—figure supplement 2C phylogeny and I didn't see any methods for alignment or tree construction either. The ECH-containing orthogroup on PLAZA (https://bioinformatics.psb.ugent.be/plaza/versions/plaza_v4_dicots/) contain several other ECH homologs (Solyc07g043690.1, Solyc07g044730.2, Solyc07g044710.1, Solyc07g044720.1) that appear to be grouping between AECH1 and Solyc06g54520, but these sequences are not included in the supplemental phylogeny.

Methods for building the phylogenetic tree in Figure 4—figure supplement 2 was added to the figure legend. The purpose of this tree is to assist analyzing the ECH enzyme functions and thus did not include the genes with no trichome expression. A detailed ECH gene phylogenetic analysis with Sl-AECH1 homologs from multiple species was performed as shown in Figure 4—figure supplement 3.

2) Framing2a) The authors mention in the Abstract and again in the Introduction that specialized metabolic gene clusters (SMGCs) are a hallmark of plant specialized metabolism. This gives the impression that gene clusters in plants are more common than they are. In fungal and bacterial genomes, gene clusters predominate. In plants, the pattern is not nearly as straight forward. Many plant specialized pathways are not clustered, and those pathways that are clustered are only partially so (i.e. these clusters are missing pathway regulators, product transporters, additional accessory enzymes, etc., which are common in microbial clusters); see Wisecaver et al., 2017. Given the extensive literature on complex microbial SMGCs, referring to two co-located genes as a SMGC feels like a bit of a stretch without additional clarification. Sometimes the authors appear to include the BAHD in the gene cluster, sometimes it's referred to as a two gene cluster. I wasn't sure whether BAHD was perhaps already know to function earlier in the pathway? The gene wasn't differentially expressed in the trichomes, so is it not involved? If no, then why is it included in the model?

1) We improved the way in which the concept of plant specialized metabolic gene clusters (SMGCs) was introduced to avoid giving readers the impression that plant gene clusters are common. The reference, Wisecaver et al., 2017, was cited in the introduction to provide a broader view of SMGCs prevalence in plants.

2) Additional clarification was added to the main text when describing the acylsugar gene cluster, which should be viewed in the broader context of synteny. Other than the co-localized Sl-AACS1 and Sl-AECH1 on chromosome 7, the chromosome 12 syntenic region contains the Sl-ASAT1 BAHD acyltransferase, which catalyzes the first step of tomato acylsugar biosynthesis. Notably, another acylsucrose BAHD acyltransferase, *PaxASAT2*, was also found in the *Petunia axillaris* synteny.

3) It is necessary to include BAHD acyltransferase genes in the evolutionary analysis to better understand the emergence of the tomato acylsugar cluster, especially because the presence of BAHD acyltransferase in the synteny predates the divergence of Solanaceae and Rubiaceae (refer to Figure 6 and supplements).

2b) Similarly, I question the use of the term supercluster here as well. The fumagillin and pseurotin superclusters in Aspergillus are at least 29 genes long, co-regulated, intertwined, and maintained in diverse Aspergillus species despite being in the telomeric region of otherwise rapidly rearranging genomes (Wiemann et al., 2013). Do the authors believe that the co-location of these genes is being maintained in Solanum and are the genes being co-regulated?

We added the reference raised by the reviewer and added more discussion of possible causes for the phenomenon and less on the terminology. As it is the case that plant clusters are rare, we expect plant superclusters to be even rarer and almost certainly smaller those in fungi. To answer the question of coregulation of the Chr 7/12 synteny genes requires future work.

Reviewer #2:[…]1) Novelty and framework for discussing innovation in metabolism with respect to clusters and “superclusters.” The novelty of the manuscript (for researchers who work outside Solanaceae and trichome biology) lies in the final paragraph discussing how this may be the first paper to describe "superclusters." The Abstract, Introduction and Discussion could all benefit by citing more literature that encompasses the ongoing debates about the prevalence of "clusters" or not. The authors ask this as a question in the final paragraph; however this has been addressed in research not cited here (for example, Wisecaver et al., 2017). As the paper currently reads, they leave off most research (aside from their own self citations) that is outside of the "cluster paradigm" of the Osbourn lab (e.g., Nutzmann et al.2016). It would be interesting to know if a "Wisecaver-type network analysis" would get similar results.

We revised the Abstract, Introduction, and Discussion to provide a broader view of the clustered and non-clustered specialized metabolism (SM) genes. Wisecaver et al., 2017 proposed that plant SM pathway genes are co-expressed, independently of being organized into biosynthetic gene clusters. To test whether Wisecaver’s method will uncover which tomato genes in the chromosome 7 synteny are acylsugar biosynthesis related, we performed co-expression analysis using ACS, ECH, and BAHD acyltransferase family genes (Figure 2—figure supplement 1 and Supplementary file 1). As expected, the genes that were characterized involved in acylsugar biosynthesis (Sl-AACS1, Sl-AECH1, and four Sl-ASATs) grouped together. One hypothesis for both physical proximity and gene expression clustering of Sl-AACS1 and Sl-AECH1 is that gene colocalization increases the possibility of coregulation. For example, a promoter element could be duplicated/transposed to drive the expression of nearby genes or newly inserted genes. Searching for the promoter element and testing our hypothesis is an on-going research in the lab.

2) Gene and genome duplication analyses. The authors discuss evolution and gene duplication; however, they do not use the genomic resources of the 13 species examined to phylogenetically evaluate whether the duplicates that may be involved in the phenotype are from whole genome duplication (WGD or polyploidy) vs. small scale duplication (SSD). Given that Solanaceae have a WGD (as a triplication shown in Figure 6A), it would seem important to investigate the acylsugar biosynthesis pathway in this context (and any inferred patterns of loss after triplication if now in single or duplicate copy).

Please see our response to a similar question from reviewer 1.

3) Experimental Design. A species phylogeny is given in Figure 6B for the species analyzed and the corresponding results and discussion hypothesize gain and loss events. Do the authors use a formal ancestral reconstruction analysis to back up their evolutionary scenario or is it just using parsimony to explain biochemical observations? There are formal tools used by evolutionary biologists to infer trait evolution that could be used here. This would seem particularly relevant for any claims made about the other genera (Jaltomata, Physalis, Iochroma, Atropa, and Hyoscyamus) that have missing data. It is not required; however, there other resources for other genera and sister families that could be exploited in a transcriptome analysis (e.g. One Thousand Plant Transcriptomes Initiative. 2019. One thousand plant transcriptomes and phylogenomics of green plants. Nature 574: 679-685).

1) We performed trait ancestral state reconstruction analysis using the likelihood method instead of parsimony as shown in (Figure 6—figure supplement 7). Four traits were inferred for their ancestral states. They are the presence of medium chain acylsugars, presence of ACS in the synteny, presence of ECH in the synteny, and presence of both ACS and ECH in the synteny. We used the ancestral state reconstruction to infer that the co-emergence of the medium chain acylsugars and the ACS/ECH genes in synteny occurred in the common ancestor of *Solanum*.

2) Due to the highly trichome-specific expression patterns of genes involved in acylsugar biosynthesis, transcriptome analysis using tissues such as leaves, roots, fruits etc. does not always contain acylsugar related genes. Therefore, we refrained from using the public plant transcriptome databases to explore or validate genes involved in acylsugar biosynthesis, which may lead to biased results.

Reviewer #3:This manuscript describes investigation of biosynthesis of medium chain length acylsugars in the Solanaceae, involving identification of an intriguing biosynthetic gene cluster. In particular, two new enzymes are identified and characterized here, which were found to be part of a complex genetic loci containing three different types of enzymes, each of which has undergone multiple tandem gene duplication. However, those for the “other” (third) enzyme in the original (chromosome 7) cluster are not required for this medium chain acylsugar biosynthesis, but rather a paralog on chromosome 12 instead. Intriguingly, this cluster is adjacent to the larger portion of the split cluster previously identified for steroidal alkaloid biosynthesis. It is tempting to speculate that assembly, with subsequent splitting, occurred together. Although the authors have avoided this more speculative hypothesis, it might be of interest to look at conservation of the split steroidal alkaloid cluster, analogous to that already reported here for the acylsugar biosynthetic cluster, to examine this hypothesis.

We appreciate that the reviewer provided the tempting hypothesis regarding the evolutionary history of the two metabolic gene clusters. The analysis of co-location or co-regulation of the genes involved in producing these two compounds in the *Solanum* genus and beyond is an on-going interest of the lab.

In addition, one of the biochemical findings reported here is somewhat puzzling. Specifically, the characterized activity of the acylsugar acyl-CoA synthetase (AACS) seems to be higher with short, particularly C8, rather than medium (>C10) chain length fatty acids. The authors should at least note this in the Discussion. Otherwise the work is solid, the results interesting and well-presented.

We mentioned and discussed this point that Sl-AACS1 seems to have higher activity with C8 fatty acids. Our favorite (though not experimentally tested) hypothesis for lack of C8-containing acylsugars is that this reflects a dearth of C8 fatty acids in trichomes.

[Editors' note: further revisions were suggested prior to acceptance, as described below.]

Reviewer #1:This revision satisfies most of my comments on the previous draft. The evolutionary analysis of the three gene families is much improved.Could the segmental duplication of acyl-CoA have occurred in an ancestor of Solanum and Nicotiana following the divergence from Petunia? That seems like the most parsimonious explanation. What am I missing?

It is unlikely that the segmental duplication of acyl-CoA synthetase occurred in an ancestor of *Solanum* and *Nicotiana* following the divergence from *Petunia*. The rationale is as follows.

In Figure 6—figure supplement 4, there are two clades that are relevant to this:

- The first is the dark blue branch containing SI-AACS1 (call this Clade A).

- The other is the cyan branch containing Solyc02g082880 (call this Clade B).

The mechanism of the duplication event leading to clade A/B (labeled (4) in the tree) is what we are interested in. Consider that:

- Clade A has no *Petunia* homolog, this does not inform our understanding of whether the duplication (4) took place before or after.

- Clade B on the other hand contains *Petunia* homologs.

Based on the two lines of information above, one of the most parsimonious explanations is that:

- A duplication event took place before the *Petunia* and the tomato/tobacco lineages split, and

- A loss event occurred in the *Petunia* lineage, leading to the absence of the *Petunia* gene in clade A.

While we agree that the reviewer’s hypothesis is parsimonious, we suggest it to be unlikely. This is because it would require a gene gain event; for example, horizontal gene transfer or introgression in *Petunia* in addition to the segmental gene duplication event that the reviewer suggested. We hope that you will agree that a gene gain event in *Petunia* is of lower probability than our proposed gene loss event.

In the response to my earlier comment, the authors say that understanding of the mechanism of hijacking primary metabolism for acylsugar biosynthesis requires future work. However, the manuscript still reports to provide insights into this co-option in several places. Can the authors tone down these statements or make it more explicit what they mean here?

In the Abstract, we edited the last sentence to “This work reveals insights into the dynamics behind gene cluster evolution and cell-type specific metabolite diversity.” We also changed the last sentence of the Introduction to “These results provide insights into specialized metabolic evolution through emergence of cell-type specific gene expression, the formation of metabolic gene clusters and illuminates additional examples of primary metabolic enzymes being co-opted into specialized metabolism.”

I think the point of the co-expression analysis was missed a bit by focusing only on other ACS, ECH, and BAHDs. It looks like the Moore et al., 2020 study called modules of co-expressed genes using several different metrics, while here the authors only report the pairwise coexpression between ACS, ECH, and BAHDs without binning these genes into discrete modules or looking for associations with other gene families. This may be beyond the scope of this manuscript. However, I think it is definitely something worth investigating at some point to better understand the evolution of the larger metabolic pathway rather than the portion contained within this genomic region.

We agree that a more complete analysis is worth investigating in the future.

Reviewer #2:[…]The only substantial issue is that is it not clear to me what a "super-cluster" is or why that new terminology is needed here. The response to reviewers was clearer than the revised manuscript about this issue (relative to what is seen in fungi as noted by another reviewer).

We removed the terminology “supercluster” from the manuscript to avoid confusion with the fungal paradigm.